# The ULK3 kinase is a determinant of keratinocyte self-renewal and tumorigenesis targeting the arginine methylome

Sandro Goruppi [1,2] ✉, Andrea Clocchiatti[1,2], Giulia Bottoni [1,2], Emery Di Cicco[1,2], Min Ma[3,4], Beatrice Tassone[3,4], Victor Neel[2], Shadmehr Demehri [1,2], Christian Simon[3,4,5] & G. Paolo Dotto [1,2,3,4,5] ✉

Epigenetic mechanisms oversee epidermal homeostasis and oncogenesis. The identification of kinases controlling these processes has direct therapeutic implications. We show that ULK3 is a nuclear kinase with elevated expression levels in squamous cell carcinomas (SCCs) arising in multiple body sites, including skin and Head/Neck. ULK3 loss by gene silencing or deletion reduces proliferation and clonogenicity of human keratinocytes and SCC-derived cells and affects transcription impinging on stem cell-related and metabolism programs. Mechanistically, ULK3 directly binds and regulates the activity of two histone arginine methyltransferases, PRMT1 and PRMT5 (PRMT1/5), with ULK3 loss compromising PRMT1/5 chromatin association to specific genes and overall methylation of histone H4, a shared target of these enzymes. These findings are of translational significance, as downmodulating ULK3 by RNA interference or locked antisense nucleic acids (LNAs) blunts the proliferation and tumorigenic potential of SCC cells and promotes differentiation in two orthotopic models of skin cancer.

Squamous cell carcinoma (SCCs) is among the most frequent types of human cancer and represents a significant cause of death[1]. These heterogeneous tumors originate from surface epithelial cells, such as the skin, aerodigestive and urogenital tracts, in response to various noxious conditions[2,3]. Two main categories of gene alterations, driver mutations and more selective mutations that impact squamous cell fate commitment and/or ensuing terminal differentiation, are commonly found in these tumors[3].

NOTCH signaling is a crucial determinant of keratinocyte (KC) proliferation and differentiation[3]. An inverse regulatory relationship exists between NOTCH and TP63 activity, the latter promoting KC self-renewal potential and early steps of tumorigenesis by acting on other key regulators of KC proliferation, such as p21/CDKN1A and MYC[4–6].

This interplay is similarly important in oral, esophageal[7], bronchial[8], and cervical[9] epithelia.

Another critical signaling with oncogenic potential for KCs is the Hedgehog (Hh)/GLI pathway[10,11]. Mammalian cells express three GLI (GLI1-3) transcription factors (TFs), with *GLI2* having a role in KC oncogenic conversion, not only in basal cell carcinomas (BCCs) but also SCCs[11,12]. While increased Hh signaling by *patched* (*PTCH*) and *smoothened* (*SMO*) gene mutations were causally implicated in BCCs[13], the mechanisms of GLI2 activation in the SCCs remain to be clarified.

The Unc51-like kinase 3 (ULK3) is a poorly characterized member of the ULK1-4 kinase family proposed to fulfill in mammalian cells the same central role as the Drosophila fused kinase towards GLI2

[1]Cutaneous Biology Research Center, Massachusetts General Hospital, Charlestown 02129 MA, USA. [2]Department of Dermatology, Massachusetts General Hospital and Harvard Medical School, Boston 02114 MA, USA. [3]Personalized Cancer Prevention Research Unit and Head and Neck Surgery Division, Centre Hospitalier Universitaire Vaudois, Lausanne 1011, Switzerland. [4]Department of Immunobiology, University of Lausanne, Epalinges 1066, Switzerland. [5]International Cancer Prevention Institute, Epalinges 1066, Switzerland. ✉e-mail: sgoruppi@mgh.harvard.edu; paolo.dotto@unil.ch

activation[14,15]. This kinase has been mainly studied for its cytoplasmic functions in breast cancer and muscle cells[14,16,17] and implicated in the endosomal sorting machinery by delaying the abscission of midbodies under stress conditions in HeLa cells[17]. We and others have shown that ULK3 physically associates with the GLI2 protein and enhances its transcriptional activity, as indicated by the expression of its direct target gene, *GLI1*, and *GLI2* itself[15,18,19]. In the stroma of tumors, including SCCs, ULK3 is critical for the conversion of normal fibroblasts to activated cancer-associated fibroblasts[18]. By contrast, the function of ULK3 in the epithelial compartment of tumors is unknown.

"*Transcriptional addiction*" of cancer cells can result from altered cross-talk between ubiquitous and cell type-specific TFs[20]. From this perspective, TFs can be divided into three functional classes[20]: determinants of cell fate (such as TP63), integrators of various developmental and cell signaling pathways (such as NOTCH/CSL and GLI) and general regulators of proliferation (such as MYC, AP1, and p53). The TFs of these classes all bind and regulate *super-enhancers (SEs)*, open chromatin regions with a high H3K27 acetylation density[21]. Many chromatin writers, including the histone acetyltransferase and transcriptional coactivator EP300, are recruited to such regions via direct interaction with TFs such as TP63 and MYC[22,23]. Dramatic changes in *SE* organization that enhance binding sites for a cohort of TFs, including TP63, have been reported in SCCs[24] and are a hallmark of cancer[20].

Altered metabolism is also a hallmark of cancer growth. Several TFs, such as c-MYC and FOXM1, oversee cellular glycolytic metabolism by directly regulating the expression of rate-limiting enzymes and transporters, such as PKM, LDHA, PDKs and GLUT1[25–27]. Lung, Head/Neck, and cutaneous SCCs utilize glucose metabolism to meet their anabolic/catabolic needs for rapid growth and can adapt by using alternative amino acids such as glutamine as a source of energy to circumvent the inhibition of glycolysis[28–30]. This highlights the need to identify druggable targets and develop metabolism-targeting therapies for cancer.

The arginine methylation of protein residues is key for many cellular processes, including epigenetic modifications, and affects the control of proliferation, stem cell potential, and carcinogenesis[31]. There are two types of protein arginine methyltransferases (PRMTs), those that catalyze asymmetric (type I) versus symmetric (type II) dimethylation. PRMT1 is the best-studied type I enzyme and mainly exhibits transcription-activating functions, while PRMT5 is the best-studied type II enzyme, and is endowed mostly with transcription-repressive functions[32]. Despite reported opposite roles in some contexts, PRMT1 and PRMT5 have a shared transcription regulatory function as part of the arginine methylome complex associated with DNA[33]. These PRMTs regulate the epigenetic modification of histone four arginine 3 (H4R3), which can impact the recruitment of several writers, including EP300, to chromatin[34,35]. PRMT1 and PRMT5 are downregulated in differentiating keratinocytes, and sustained PRMT1 expression was found to suppress the differentiation of these cells[36].

We report here that ULK3 is a determinant of the self-renewal and oncogenic potential of human primary keratinocytes (HKCs) and SCC cells, a nuclear kinase that impinges on TP63, GLI, and metabolic transcription regulatory pathways. ULK3 binds directly and controls the function of PRMT1 and PRMT5 in chromatin modification and gene expression. The targeting of this nuclear kinase can represent a strategy for treating cancers, specifically SCCs.

## Results

### Clinical SCC samples and SCC-derived cell lines have elevated ULK3 levels

A dynamic KC-specific network of epigenetic modifications and TFs are involved in squamous cell fate determination and oncogenesis[2,3]. Identifying and targeting nuclear kinases in this network constitute a novel strategy for cancer therapy[37]. Analysis of large-scale transcriptomic datasets showed consistently higher levels of *ULK3* gene expression in SCCs arising in multiple body sites (Fig. 1a). These findings were extended by immunofluorescence analysis of tissue arrays of skin, head and neck and cervical cancer SCCs, which showed consistent and statistically significant increases in ULK3 levels in tumor samples versus normal or nonaffected adjacent tissues (Fig. 1b). A more detailed immunofluorescence analysis of cutaneous SCCs showed prominent ULK3 nuclear expression in a large fraction of skin SCC cells with little expression in keratinocytes of normal skin (Fig. 1c and Supplementary Fig. 1a). ULK3 upregulation in SCCs was coincidental with the expression of the Ki67 or TP63 proliferation markers, as determined by double immunofluorescence with the corresponding antibodies (Fig. 1d). Actinic keratoses (AK) are premalignant keratinocyte lesions associated with an altered squamous cell differentiation program and expansion of cells of the proliferative compartment[38]. As in SCCs, immunofluorescence analysis of these lesions showed an accumulation of nuclear ULK3 levels (Fig. 1e and Supplementary Fig. 1b), with co-localization with TP63 (Fig. 1f and Supplementary Fig. 1c).

We further evaluated *ULK3* expression in a panel of SCC cell lines derived from oral and skin lesions and compared them to those in several strains of primary HKCs. Among the four ULK family members, *ULK3* was the only one that was consistently upregulated (Fig. 1g and Supplementary Fig. 1d). ULK3 protein levels were also significantly upregulated in SCC cells versus primary HKCs, as assessed by immunoblot assays with two different anti ULK3 antibodies (Fig. 1h).

Thus, elevated ULK3 expression is a consistent hallmark of SCCs arising at various body sites, which is also observed in premalignant AK lesions and maintained in cultured SCC cells relative to normal HKCs.

### ULK3 downregulation or inhibition suppresses the proliferative potential of HKCs and SCC cells

To assess the role of *ULK3* in HKCs and SCC cells, we started by infecting these cells with two *ULK3*-silencing lentiviruses versus an empty control vector. Effective downregulation of *ULK3* expression in both HKCs and SCC cells (Supplementary Fig. 2a, b) resulted in significant suppression of cell proliferation (Fig. 2a) with increased fractions of apoptotic and necrotic cells (Fig. 2b). The colony-forming capability, as a well-established stem/self-renewal assay[39], of HKCs, of skin and Head/Neck SCC cells and cervical cancer HeLa cells was also significantly reduced by shRNA-mediated *ULK3* silencing (Fig. 2c and Supplementary Fig. 2c).

As an alternative approach, we resorted to *ULK3* gene deletion by CRISPR/Cas9 technology[40]. To eliminate the issue of individual clone variability and possible compensatory mutations resulting from *ULK3* loss, we analyzed pooled polyclonal cell populations shortly selected after infection with lentiviruses co-expressing Cas9 together with *ULK3*-targeting guide RNAs (gRNAs). Infection of SCC13 cells with lentiviruses expressing three different *ULK3* gRNAs resulted in the effective deletion of the gene, as assessed by Surveyor assays[41], with PCR amplification of the *ULK3* gene-targeted regions followed by nuclease digestion of mismatched heteroduplexes (Fig. 2d). Deletion of the *ULK3* gene was accompanied by loss of ULK3 protein expression (Fig. 2e) and, as shown in Fig. 2f, a marked decrease of SCC cells colony-forming capability. A similar reduction of clonogenicity and proliferative activity was observed after treatment of SCC cells with SU6668, an inhibitor of ULK3 kinase activity[19] (Supplementary Fig. 2d–f).

Molecularly, *ULK3* gene silencing resulted in the strong downregulation of TP63, a positive determinant of HKC proliferation and self-renewal[42] (Fig. 3a, b), with a similar reduction of TP63 levels occurring in SCC13 cells treated with the SU6668 inhibitor[43] (Supplementary Fig. 3). In agreement with its reported function in other cellular systems[16,18], ULK3 loss also suppressed expression of the *GLI1/2* transcription factors, which play a positive role not only in BCCs but also SCCs[12] (Fig. 3c). Conversely, *ULK3* silencing resulted in the upregulation of keratin 1 and 10 (*KRT1*, *KRT10*), involucrin (*IVL*) and filaggrin (*FLG*) differentiation markers (Fig. 3d, e) and induction of *CDKN1A*/p21,

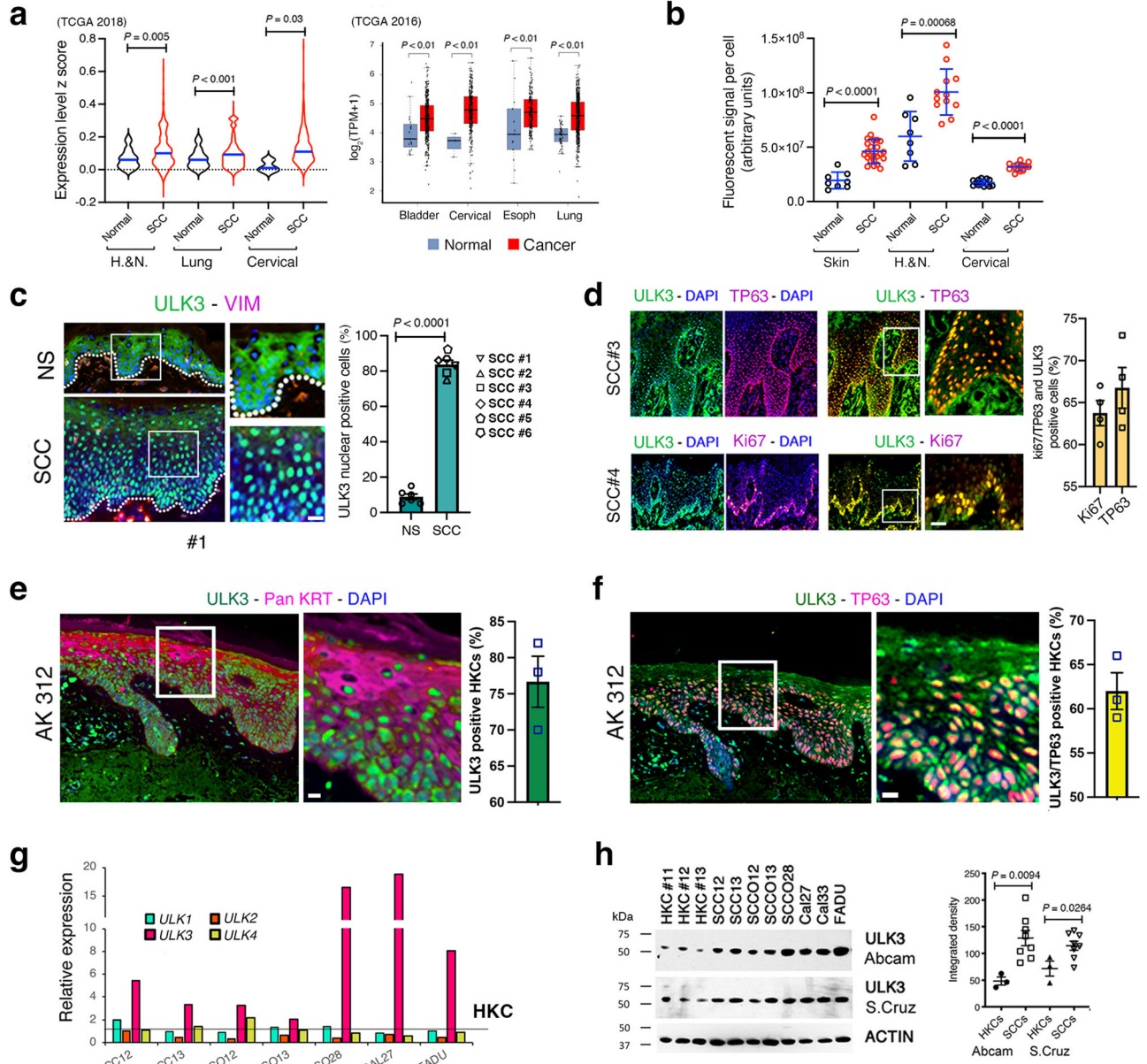

**Fig. 1 | ULK3 is upregulated in clinical SCC samples and SCC cells. a** *ULK3* expression in TCGA profiles of Head/Neck (H&N), Lung, and Cervical Squamous Cell Carcinomas (SCCs) versus matched normal tissues updated to 2018. Individual z-scores values, mean ± SEM, *P* < 0.01, two-tailed unpaired t-test. Box plots show *ULK3* levels in the indicated SCCs versus normal tissues in TCGA datasets updated to 2016 as in ref. [2]. GEPIA-generated median, box (25–75%) and whiskers (5–95%), log(transcript per million+1), value cutoff *P* < 0.01. **b** Quantification of Skin, H&N and Cervical SCCs tissue arrays ULK3 immunostaining with unmatched (Skin and H&N) or matched (Cervical) non-affected tissues. Skin: *n*(Normal = 8, SCC = 24); H&N: *n*(Normal = 8, SCC = 26); Cervical: *n*(Normal = 14, SCC = 14), mean ± SEM, *P* < 0.001, two-tailed unpaired *t*-test. **c** ULK3 (green) and VIMENTIN (magenta) immunofluorescence and quantification of nuclear ULK3 in normal skin (NS) and skin SCC. Representative low and high magnifications, *n*(NSs/SCCs) = 6, mean ± SEM, *P* < 0.0001, two-tailed unpaired t-test. Scale bar 10 μm. ULK3 immunofluorescence of additional NSs/SCCs is in Supplementary Fig. 1a. **d** ULK3 (green) and Ki67 (magenta) or TP63 (magenta) immunofuorescence of skin SCCs (SCC#3 and SCC#4). Representative low and high magnifications with quantification of double nuclear ULK3/TP63 (yellow) or ULK3/Ki67 (yellow) positivity. *n*(SCCs) = 4, mean ± SEM, Scale bar 20 μm. **e** ULK3 (green) and pankeratin (magenta) immunostaining with quantification of ULK3 in actinic keratosis lesions (AKs). Representative low and high magnifications, *n*(AKs) = 3, mean ± SEM. Scale bar 5 μm. **f** ULK3 (green) and TP63 (magenta) immunostaining with quantification of ULK3/ TP63 positive cells (yellow) in AKs. Representative low and high magnifications, *n*(AKs) = 3, mean ± SEM. Scale Bar 5 μm. Additional AK ULK3/TP63 immunostaining is in Supplementary Fig. 1b, c. **g** RT-qPCR analysis of *ULK* family members (*ULK1-4*), normalized to *36β4*, in SCC cells versus average level (solid line) in human keratinocytes (HKCs) (strains #GB2- #GB10). *n*(HKCs) = 9, *n*(SCCs) = 7. Representative experiment of two independent biological replicates. RT-qPCR of *ULK1-4* in each HKCs and SCCs are in Supplementary Fig. 1d. **h** Immunoblot and densitometric quantification of ULK3 in HKCs (#GB11- #GB13) and SCC cells (normalized to ß-ACTIN). *n*(HKCs) = 3, *n*(SCCs) = 8, mean ± SEM, *P* < 0.05, two-tailed unpaired *t*-test. *The researcher's initials and a number identify each primary HKC strain. A number identifies patient-derived SCC samples.*

a key mediator of differentiation-associated cell cycle withdrawal[44] (Fig. 3f, g).

To address the consequences of increased *ULK3* expression, we infected SCC cells with an *ULK3* overexpressing lentivirus versus empty vector control. Increased *ULK3* expression significantly enhanced the colony-forming capability of SCC cells with a concomitant enhancement of sphere formation as an alternative assay of cell expansion potential[39] (Fig. 3h, i), which were paralleled by up-

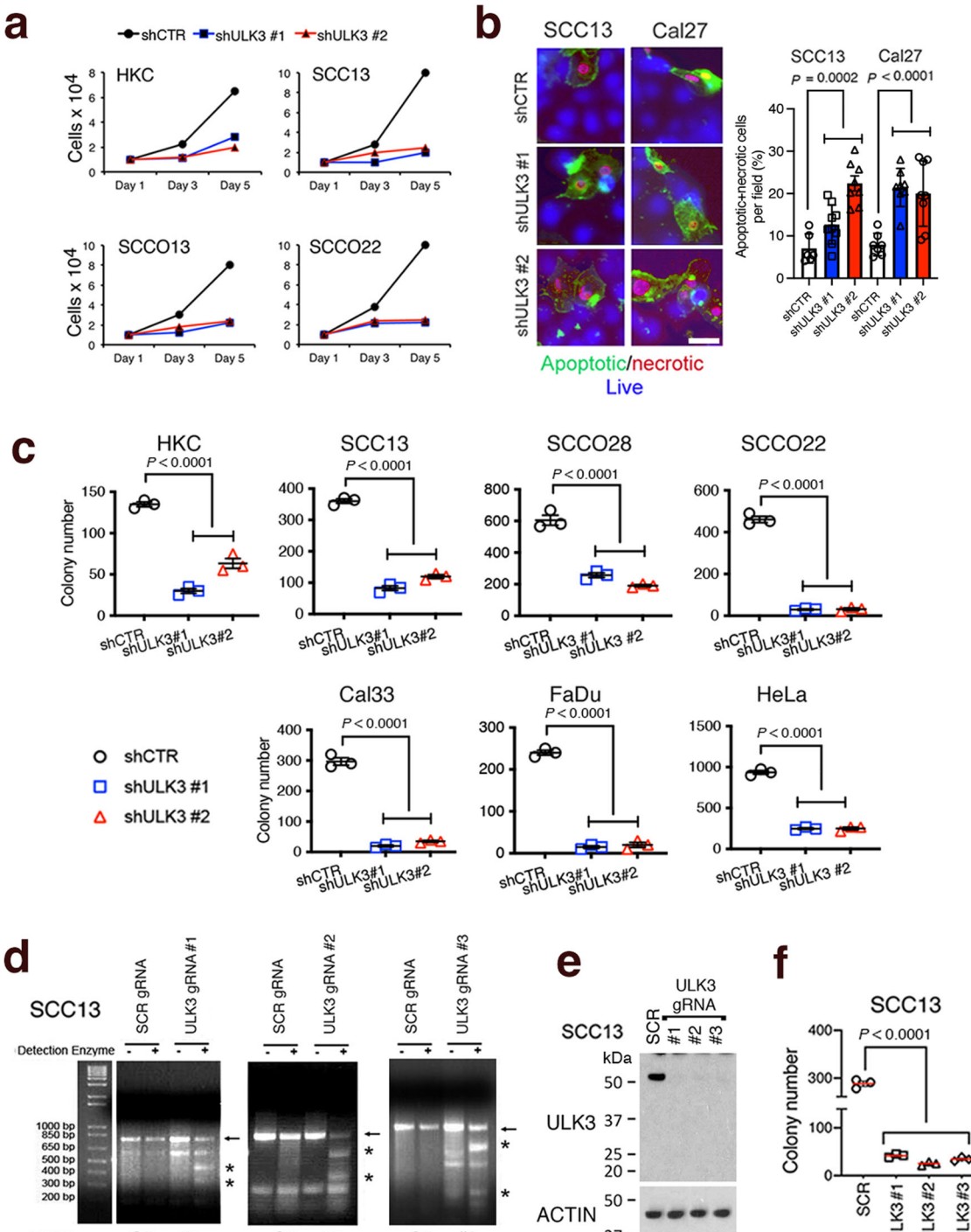

**Fig. 2 | The proliferative potential of HKCs and SCC cells is suppressed by ULK3 downmodulation. a** Cell growth on days 3 and 5 of HKC (strain #TP26) and of three SCC cell lines infected with two *ULK3* silencing lentiviruses (shULK3 #1, #2) versus empty control vector (shCTR) plated in triplicate at low-density. Representative experiment of two independent biological replicates. The efficient downregulation of *ULK3* in HKCs and SCC cells is in Supplementary Fig. 2a, b. **b** Live fluorescence images of SCC13 and Cal27 cells infected with two *ULK3* silencing lentiviruses, or control virus, and stained for early- (phosphatidylserine-FITC, green), late- apoptosis (7-aminoactinomycin D, magenta) and for all viable cells (CytoCalcein, violet-blue). Quantification, in duplicate cultures, SCC13 *n*(fields/shCTR) = 6, *n*(fields/shULK3) = 25; Cal27 *n*(fields/shCTR) = 7, *n*(fields/shULK3) = 24, *n*(cells/field) > 90, mean ± SEM, *P* < 0.001 or *P* < 0.0001 one-way ANOVA. Scale bar 10 μm. **c** Colony formation of HKCs (strain # TP26), five SCC cell lines and HeLa cells infected with two *ULK3* silencing lentiviruses, or a control virus, and plated at limited cell density for one week. Crystal violet stained dishes were quantified using ImageJ.

*n*(dishes) = 3, mean ± SEM, *P* < 0.0001, one-way ANOVA. Representative images of the stained colonies are in Supplementary Fig. 2c. Supplementary Fig. 2d-e show similar assays with SCC cells treated with the ULK3 inhibitor SU6668[19]. **d** Surveyor Assay[41] of SCC13 cells one week after the infection with lentiviruses co-expressing the Cas9 enzyme and one of the three RNA guides targeting *ULK3* (gRNA ULK3 #1-3), or a scrambled control gRNA (SCR gRNA). Each targeted genomic region (amplified using corresponding oligo pairs (#1-3) (-)) was tested for DNA mismatch by PCR by incubation with single-strand DNAse. (+). In control reactions, we used genomic DNA from SCR-infected SCC13 cells. The arrows indicate the amplified region and the asterisks the DNAase cleavage products. **e** ULK3 and ß-ACTIN immunoblotting of SCC13 cells one week after the infection with *ULK3* gRNA (#1-3), versus a scrambled gRNA (SCR) lentiviruses. **f** Colony formation of SCC13 cells, plus/minus *ULK3* deletion as above, plated at limited dilution for six days. *n*(dishes) = 3, mean ± SEM, *P* < 0.0001, one-way ANOVA.

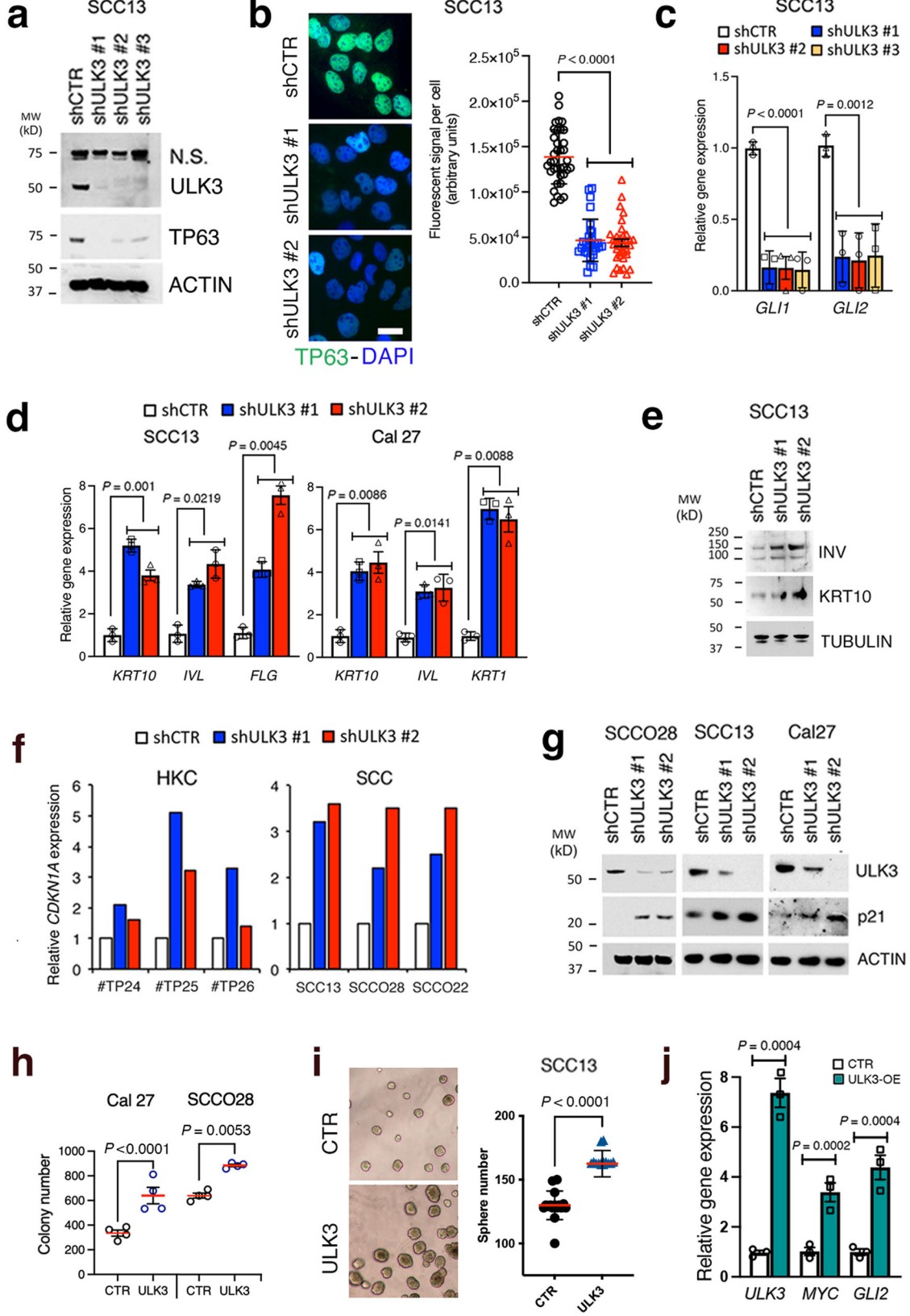

regulated expression of pro-proliferation genes such as c-*MYC* and *GLI2* (Fig. 3j).

Thus, ULK3 is a key determinant of HKC and SCC cell proliferative potential that affects other critical regulators of the growth/differentiation of these cells.

## ULK3 controls gene expression and cellular metabolism

For further mechanistic insights, we performed a global transcriptomic analysis of three HKC strains in which the *ULK3* gene was silenced by infection with two different lentiviruses versus empty vector control. As shown in Fig. 4a, several hundred genes were up or

**Fig. 3 | ULK3 silencing downmodulates proliferation/stem markers while upregulating differentiation genes and cell cycle inhibitor p21. a** ULK3, TP63 and ß−ACTIN immunoblotting of SCC13 cells infected with three *ULK3* silencing lentiviruses (shULK3#1-3), or an empty vector (shCTR), for one week. **b** TP63 (green) immunostaining and quantification of fluorescence signal intensity of SCC13 cells infected with two *ULK3*-silencing (shULK3 #1 and #2) versus a control (shCTR) lentivirus for one week. Individual cell values and mean ± SD, *n*(cells) > 40, *P* < 0.0001, one-way ANOVA. Scale bar 7 μm. **c** RT-qPCR analysis of *GLI1* and *GLI2* expression, normalized to *36β4*, in SCC13 cells plus/minus infection with *ULK3* silencing or control lentiviruses as in panel (**a**). *n*(dishes) = 3, mean ± SEM, *P* < 0.01, one-way ANOVA. **d** RT-qPCR analysis of the indicated genes, normalized to *36β4*, in SCC cells infected with two *ULK3* silencing lentiviruses or a control virus. *n*(dishes) = 3, mean ± SEM, *P* < 0.001, one-way ANOVA. **e** Keratin 10 (KRT10), INVOLUCRIN (INV) and ß−ACTIN (ACTIN) immunoblotting of SCC13 cells with two *ULK3* silencing lentiviruses for one week versus a control virus. **f** RT-qPCR analysis

of *CDKN1A*, normalized to *36β4*, of HKCs (strains # TP24, # TP25, # TP26) and three SCC cell lines plus/minus *ULK3* silencing for one week. *n*(HKCs) = 3, *n*(SCCs) = 3. **g** p21, ULK3 and ß−ACTIN immunoblotting of three SCC cell lines plus/minus *ULK3* silencing for one week. *n*(SCC lines) = 3. **h** Colony formation assays of SCC cell lines infected with a *ULK3* overexpressing lentivirus (ULK3), or an empty virus (CTR), and plated at low density for one week. *n*(dishes) = 4, mean ± SEM, *P* < 0.001, two-tailed unpaired t-test. **i** Sphere-forming capability of SCC13 cells infected with a *ULK3* overexpressing virus (ULK3) or a control virus (CTR) and cultured for one week in Matrigel suspension as in[68]. Representative phase contrast images and quantification of the number of spheres in four fields per dish. *n*(dishes/condition) = 4, mean ± SEM, *P* < 0.0001, two-tailed unpaired t-test. **j** RT-qPCR analysis of the indicated genes, normalized to *36β4*, in SCC13 cells plus/minus ULK3 overexpression (white and green respectively) as tested in panel (**i**). *n*(dishes) = 3, mean ± SEM, *P* < 0.001, two-tailed unpaired t-test.

---

downmodulated by *ULK3* silencing (GSE183084). Gene set enrichment analysis (GSEA)[45] showed a highly significant inverse correlation between the changes in gene expression elicited by *ULK3* silencing and gene signatures related to poorly versus highly differentiated Head/Neck SCCs, HKCs with silencing of the *ESR2* gene, a positive determinant of keratinocyte differentiation[46], loss of TP53, and upregulation of the pro-tumorigenic TP63, Hh/GLI and MYC signaling pathways (Fig. 4b). There was also a significant negative association between *ULK3* expression and metabolism-related gene signatures, specifically those related to cellular glycolysis and specific amino acid and nucleotide metabolism (Fig. 4b).

A similar transcriptomic analysis was performed with SCC13 cells plus/minus *ULK3* gene silencing by infection with two shRNA lentiviral vectors (GSE1830850). Gene ontology analysis showed that the genes whose expression was most significantly affected by *ULK3* silencing in both HKCs and SCC cells are involved in processes related to the cell cycle, protein translation, and metabolism (Fig. 4c). More specifically, *ULK3* silencing caused the downregulation in all cells of rate-limiting enzymes in glycolysis and glucose transport, such as *PKM, LDHA, and SLC2A1*, as well as a key TF involved in the control of metabolic genes, *FOXM1*[26,47] and the upregulation of *PDK2, PDK3* and *TIGAR*, key repressors of the glycolytic process[48,49] (Fig. 4d, e). Compromised glycolysis in cancer cells is often compensated by the increased use of amino acids as an energy source, by enhanced glutaminolysis[29]. Consistent with the suppression of glycolytic enzymes, we found that *ULK3* silencing in both HKCs and SCC13 cells resulted in the upregulation of genes coding for a glutamine-to-glutamate converting enzyme (glutaminase; *GLS1*) and glutamine-related transporters (*SLC1A5* and *SLC38A6*) (Fig. 4d). Such modulation of metabolism-related genes upon *ULK3* silencing was independently confirmed by RT-qPCR assays and immunoblot analysis of SCC cells (Fig. 4e, f).

To directly assess whether *ULK3* downregulation affects cellular metabolism, we performed a steady-state metabolomic analysis of SCC13 cells infected with two *ULK3*-silencing lentiviruses versus empty vector control. We found that ULK3 downregulation modified the levels of several metabolites involved in key cellular processes, including glycolysis, and the metabolism of lipids, nucleotides, and several amino acids, with increased levels of glutamate and glutathione, two key products of glutaminolysis[50] (Fig. 4g and Supplementary Fig. 4). Metabolite set enrichment analysis (MSEA) allows the identification of enriched metabolic pathways associated with a specific set of metabolites[51]. MSEA of SCC13 cells plus/minus *ULK3* silencing showed the most significant metabolic changes in pathways related to the metabolism of several amino acids, such as arginine/proline, homocysteine degradation and glycine/serine, methionine, and, consistently with our analysis above, glutathione and glutamate metabolism (Fig. 4h).

Thus, ULK3 downregulation impacts the expression of genes relevant for HKC self-renewal, proliferation, and metabolism, with

direct metabolic measurements consistent with decreased glycolysis and increased glutamine consumption.

## ULK3 controls the chromatin configuration of key cell regulatory genes

The prevalent nuclear localization of ULK3 in SCC cells (as shown in Fig. 1c, d and Supplementary Fig. 2b) suggested that it may associate with chromatin. To assess this possibility, we performed chromatin immunoprecipitation combined with DNA sequencing (ChIP-seq) analysis of SCC13 cells with anti-ULK3 antibodies. Global distribution of ULK3 bound peaks, determined using MACS2[52] and PeakAnalyzer[53], relative to the transcription start site (TSS) of genes in SCC13 cells showed that ULK3 associates within ±500 bp from the transcriptional start site (TSSs) of a restricted number of genes of functional significance (334) (Fig. 5a, Supplementary Fig. 5a, and GSE183933). For further mechanistic insights, we performed a ChIP-seq analysis of SCC13 cells plus/minus *ULK3* silencing with antibodies against histone 3 lysine 27 acetylation (H3K27ac), a marker of active chromatin[20,54] (GSE 183933). H3K27ac peak intensity was markedly reduced in cells with silenced *ULK3*, in areas at various distances (±1500 bp) from the TSS of genes, including most of those bound by ULK3 (Fig. 5b, c; Supplementary Fig. 5b).

Using Integrative Genomics viewer (IGV, Broad Institute) we aligned the ULK3 and H3K27ac binding peaks of two genes with key metabolic regulatory functions, pyruvate kinase M (*PKM*) and lactate dehydrogenase A (*LDHA*)[55], and a key transcriptional regulator of this process, the forkhead box M1 TF (*FOXM1*)[26,27,47]. As shown in Fig. 5d–f, upper panels, there was a broad coincidence of ULK3 and H3K27ac binding peaks at the TSS region of all three genes, with H3K27ac binding peaks being lost in cells with *ULK3* gene silencing. A similar pattern of ULK3 and H3K27ac binding was found at the TSS of the *EP300* gene, a known regulator of active chromatin and metabolism in HKCs and SCC cells [56] (Supplementary Fig. 5c).

The results were validated by independent chromatin immune precipitation (ChIP) analysis of SCC13 cells plus/minus *ULK3* silencing with two shRNA vectors. As expected, ULK3 binding to two different regions of these genes, the same as determined by ChIP-seq analysis, was markedly reduced in cells with *ULK3* silencing, with a parallel decrease of H3K27ac levels (Fig. 5d–f, middle and lower panels). The results were extended by ChIP with antibodies against histone 3 lysine 9 acetylation (H3K9ac) and RNA polymerase (Pol II), as markers of active promoters[54,57], which were also reduced in cells with silenced *ULK3* (Fig. 5d–f, lower panels). The decrease of H3K27ac, H3K9ac, and Pol II was specific for genes downmodulated by *ULK3* silencing, as it was observed with a housekeeping *TUBULIN A1A* gene, which is not bound by ULK3 nor regulated upon *ULK3* silencing in SCC13 cells (Supplementary Fig. 5d)

Thus, ULK3 oversees the chromatin configuration (open versus closed) of the majority of genes that it regulates, directly associating to

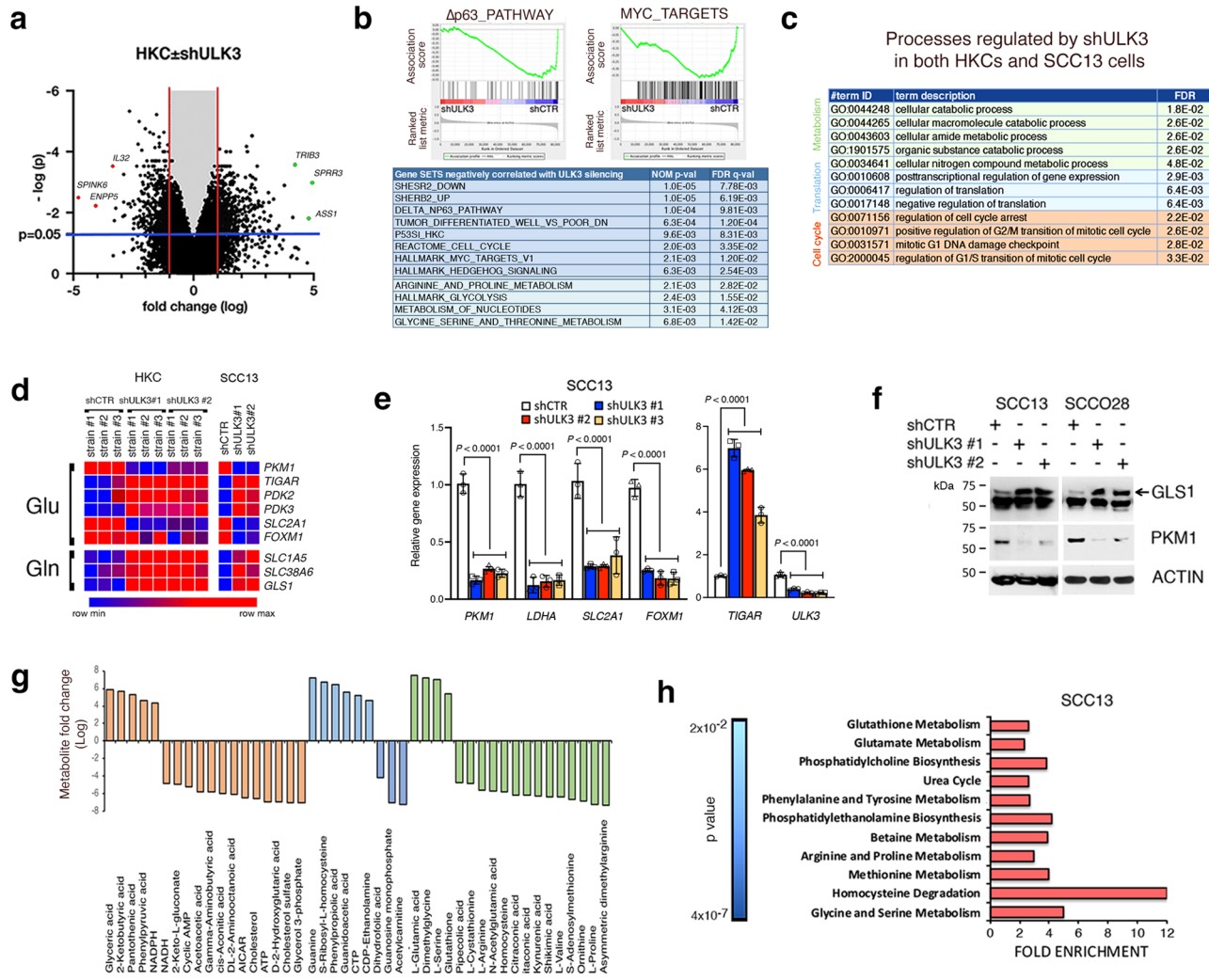

**Fig. 4 | ULK3 controls gene expression and cellular metabolism. a** Volcano plot of genes regulated in HKCs (strains #TP24, #TP25, #TP31) infected with two *ULK3*-silencing versus control lentiviruses (GSE: 183084). Marked are among most downregulated (magenta) and upregulated (green) genes. Clariom D analysis chip was made with Transcriptomic Analysis Console software (TAC) and plotted using Prism. Blue and red lines indicate the threshold of significance and ±1 fold change respectively, *n*(HKCs) = 3, *P* = 0.05, two-tailed *t*-test. **b** Gene set enrichment analysis (GSEA) of HKCs profiles with *ULK3* gene silencing versus control as in (a). Enrichment plots of ΔNp63 and MYC-related signatures, and list of most significant signatures negatively associated with profiles of *ULK3*-silenced HKCs, divided as related to growth/differentiation (blue) or cellular metabolism (green). *n*(strains) = 3, *P* < 0.001, permutation-based p values. **c** Gene ontology (GO) analysis, by database for annotation, visualization and integrated discovery (DAVID), of the processes consistently regulated by *ULK3* silencing in HKCs and SCC13 cells (GSE: 183084 and 183085), divided as pertinent to cell cycle (orange), translation (blue) and metabolism (green). **d** Heat map of expression values for components of

Glycolytic (Glu) or Glutaminolytic (Gln) pathways, from the profiles of HKCs and SCC13 cells plus/minus *ULK3* downregulation. Values visualized with Morpheus software. **e** RT-qPCR analysis of the indicated genes, normalized to *36β4*, in SCC13 cells with or without *ULK3* silencing. *n*(dishes) = 3, mean ± SEM, *P* < 0.0001, one-way ANOVA. **f** Glutaminase 1 (GLS1), Pyruvate Kinase M1 (PKM1) and ß−ACTIN immunoblots of SCC cells infected with two *ULK3* (shULK3 #1, #2) versus control silencing (shCTR) lentiviruses. The arrow indicates upregulated GLS1. **g** Steady-state metabolite profile of SCC13 infected with two *ULK3*- versus control-silencing lentiviruses (MTBL S5155). After MetaboAnalyst software[51] analysis, significantly modulated metabolites were grouped according to metabolic processes, ranked by fold changes (logFC). The values and the statistical analysis for glutathione and glutamate are in Supplementary Fig. 4. **h** Pathways affected by *ULK3* down-regulation in SCC13 cells as identified by metabolite set enrichment analysis[51]. Fold enrichment of the top 11 most significantly affected pathways ranked according to *p* values. *n*(samples/condition) = 3, *P* < 0.0, two-tailed t-test.

the regulatory regions of genes with a critical role in metabolisms and transcriptional control.

## ULK3 associates with the arginine methyltransferases PRMT1 and PRMT5

The above transcriptomic and ChIP-seq analysis indicated that only some genes are under direct ULK3 control. In contrast, the transcription and chromatin configuration of others are likely affected by *ULK3* silencing with a different mechanism. The histone arginine methylase PRMT1 plays a key role in the balance between keratinocyte self-

renewal and differentiation[36]. By GSEA, the gene signature resulting from the silencing of the *PRMT1* gene in HKCs[36] was among the most positively correlated with the transcriptomic profile elicited by *ULK3* silencing in the same cells, with individual genes of functional importance similarly modulated by *ULK3* and *PRMT1* downmodulation (Fig. 6a and Supplementary Fig. 6a, b).

An attractive possibility is that the overlapping transcriptional consequences of *ULK3* and *PRMT1* gene silencing reflect a physical association between the two proteins, which may extend to PRMT5, another PRMT family member that catalyzes symmetric rather than

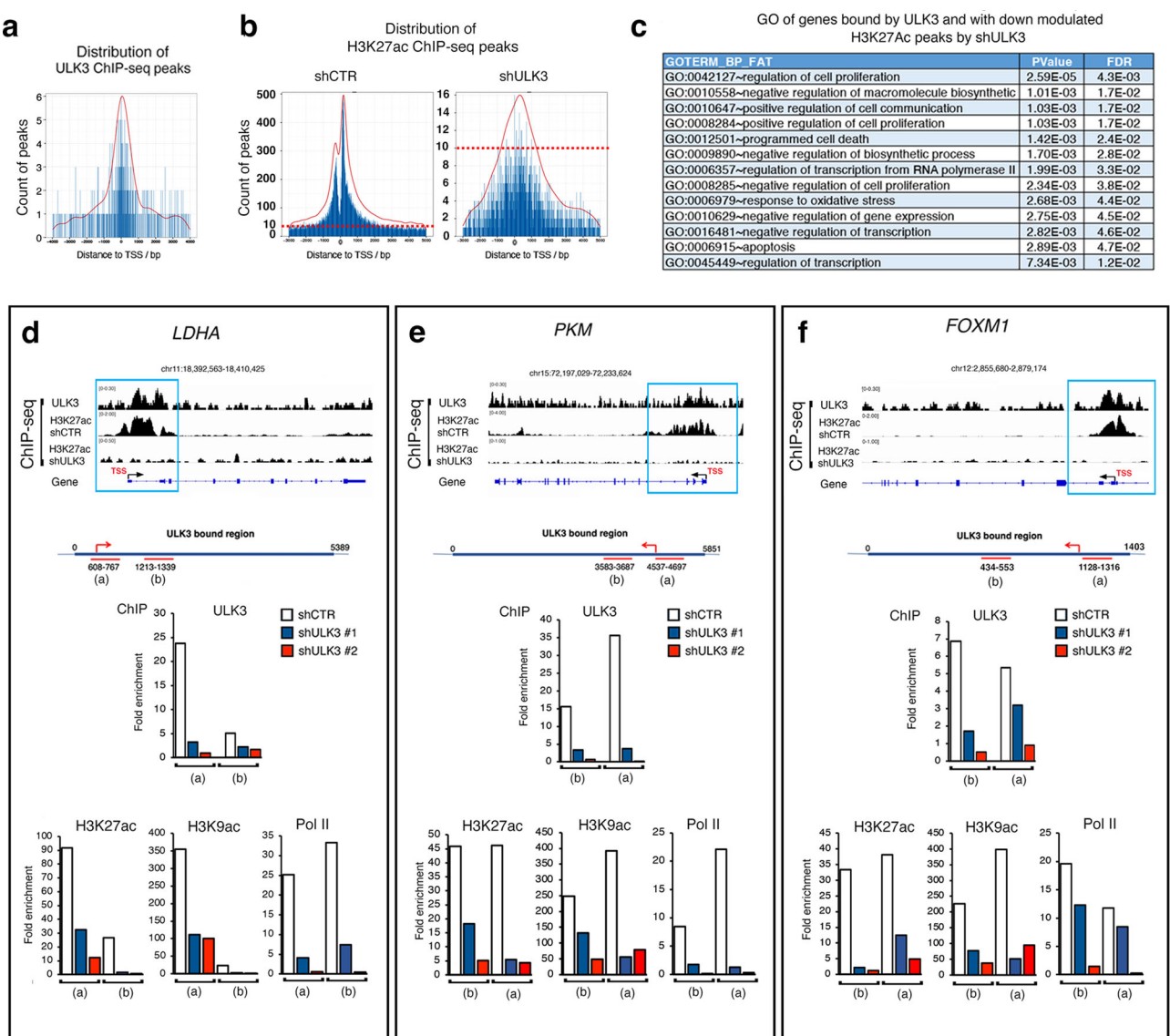

**Fig. 5 | ULK3 associates to chromatin regulatory regions of genes of functional significance. a** ULK3-bound peaks distribution as found by ChIP seq (chromatin immunoprecipitation followed by DNA sequencing) analysis of SCC13 cells with anti-ULK3 antibodies (GSE: 183933). Peaks number (count) per distance (bp) from the Transcriptional Start Site (TSS) of genes. Supplementary Fig. 5a shows the GO using DAVID software of biological processes related to the ULK3-bound genes. **b** Peaks distribution as found by ChIP-seq analysis with antibodies against acetylated lysine 27 of histone 3 (H3K27ac) in SCC13 cells plus/minus ULK3 silencing (GSE: 183933). Peaks number (count) per distance (bp) from the Transcriptional Start Site (TSS) of genes. ChIP-seq analysis was made using the Galaxy web platform. The magenta dashed lines indicate as a reference between control versus *ULK3* silenced (shRNA) cells the 10-peaks value. **c** GO using DAVID software of most significant processes of ULK3-bound genes having downregulated H3K27ac peaks upon *ULK3* silencing in SCC13 cells, as determined by ChIP seq analysis (GSE: 183933), *P* < 0.01, Fisher's exact test. Supplementary Fig. 5b shows the Venn diagram of ULK3-bound genes having downregulated H3K27ac peaks upon *ULK3* silencing. **d**–**f** Graphic illustration of ULK3- and H3K27ac -binding peaks localization on two metabolic genes (*LDHA* and *PKM*) and a transcription factor (*FOXM1*) in SCC13 cells infected with control versus *ULK3* silencing lentiviruses, aligned using the integrative genomic viewer software (IGV). Position of the TSS, exons (boxes) and introns-transcribed regions from ENCODE. Blue squares mark the main areas with H3K27ac down-modulation upon *ULK3* silencing. Supplementary Fig. 5c shows a similar analysis made on *EP300* promoter. ChIP with antibodies against the indicated proteins of SCC13 cells infected with two *ULK3* silencing (red and blue) versus control (white) lentiviruses, followed by qPCR amplification of the two regions indicated in the scheme of the *PKM*, *LDHA* and *FOXM1* genes, to which ULK3 was found to bind by CHIP-seq analysis. Results are expressed as enrichment fold relative to non-immune antibodies, *n*(experiments) = 2. Supplementary Fig. 5d provides a similar analysis for the specificity control gene *TUBA1A*.

asymmetric arginine methylation with a functional interplay with PRMT1[58,59]. Like PRMT1, PRMT5 has also been reported to play a role in keratinocyte differentiation[36,60]. Proximity ligation assays (PLA) detect endogenous intramolecular interactions, which are visualized by fluorescence puncta resulting from the juxtaposition of antibodies against associated proteins[18,61,62]. In two different SCC cell lines, Cal27 and SCC13 cells, PLA with anti ULK3 and either anti PRMT1 or anti PRMT5 antibodies generated a large number of nuclear puncta (Fig. 6b). The ULK3 -PRMT1 and -PRMT5 association was independently

confirmed by coimmunoprecipitation and immunoblotting with antibodies against the same proteins (Fig. 6c). We further investigated the interaction by coimmunoprecipitation assays in HEK293 cells transfected with expression vectors for wild-type ULK3, ULK3 with two different inactivating point mutations, (K44H), mapping within the ATP-binding site of the kinase domain[17], and (K139R), mapping in the donor/acceptor region of the kinase transferase activity[16,17], and an ULK3 truncation lacking the C-terminal region (ΔC). As shown in Fig. 6d, binding of endogenous PRMT1 and PRMT5 was equally

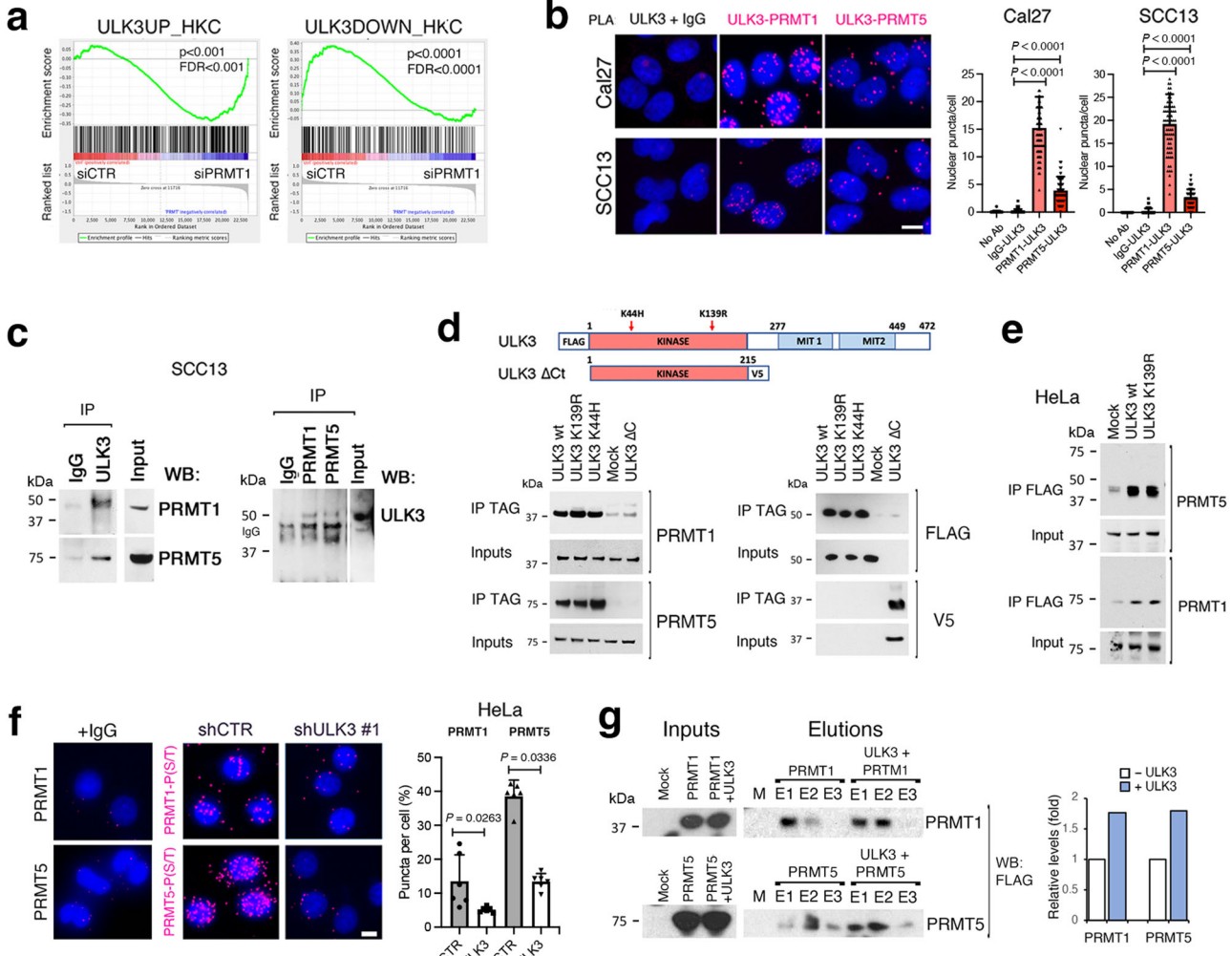

**Fig. 6 | ULK3 associates with the arginine methylases PRMT1 and PRMT5.**
**a** GSEA of HKCs profiles with PRMT1 silencing by siRNA[36] versus up- or down-regulated signatures from HKCs with *ULK3* silencing, $P < 0.001$, permutation-based p. The Venn diagram of genes modulated by ULK3 silencing and bound by ULK3 in ChIP-seq in SCC13 cells is in Supplementary Fig. 6a. The list of the genes modulated by siPRMT1 and shULK3 in HKCs is in Supplementary Fig. 6b. **b** Proximity ligation assays (PLA) of two SCC cell lines with anti -ULK3 and -PRMT1 or -PRMT5 antibodies, with anti-ULK3 alone or IgG as controls. Representative images and quantification of the magenta *puncta* from the juxtaposition of ULK3 and PRMT1 or PRMT5 antibodies, n(cells/condition) = 75, mean ± SEM, $P < 0.01$, two-tailed unpaired t-test. Scale bar 5 μm. **c** Immunoprecipitations (IP) with anti -ULK3 (ULK3) or IgG antibodies from SCC13 cells followed by PRMT1 and PRMT5 immunoblotting (left), and IPs with anti -PRMT1, -PRMT5, or IgG antibodies followed by ULK3 immuno-blotting (right). Inputs were run on the same gel. **d** ULK3 protein map with indi-cated the kinase and the microtubule-interacting (MIT1/2) domains, the positions of inactivating mutations (K139R and K44H)[16,17], C-terminus truncation (ΔC)[18] and

epitope tags (FLAG or V5). Analysis of HEK293T cells transfected with the indicated expression vectors followed by IP with anti-epitope antibodies (IP TAG) and immunoblotting with the indicated antibodies. **e** Analysis of HeLa cells mock-transfected or transfected with the indicated vectors followed by IP with FLAG antibodies (IP FLAG) and immunoblotting with the indicated antibodies. **f** PLA with anti -PRMT1 or -PRMT5 and anti -phosphorylated Ser/Thr (P-S/T) antibodies in HeLa cells infected with an *ULK3*-silencing (shULK3#1) versus control (shCTRL) lentivirus. IgGs and anti-PRMT antibodies alone served as controls. Representative images and quantification of the magenta *puncta* from the juxtaposition of the antibodies. n(fields/condition) = 6, mean ± SEM, $P < 0.01$, two-tailed unpaired t-test. Scale bar 10 μm. **g** HeLa cells were either mock transfected (M) or transfected with expression vectors for FLAG-PRMT1 or FLAG-PRMT5 alone or in combination with wild-type ULK3 (ULK3 + PRMT1 and ULK3 + PRMT5). Total extracts (Inputs) and the proteins eluted (fractions E1–E3) from a phosphoprotein-binding resin (Talon), were ana-lyzed by FLAG immunoblotting and quantified.

detected to wild-type and catalytically inactive ULK3, but not to ULK3 with the C-terminal deletion, showing that the interaction is indepen-dent of ULK3 kinase activity and maps outside the catalytic domain. Both wild-type and kinase-dead ULK3 efficiently coimmunoprecipi-tated endogenous PRMT1/5 also in HeLa cells, further validating the results (Fig. 6e).

To assess whether endogenous PRMT1 or PRMT5 phosphoryla-tion levels depend on ULK3, we performed PLA with antibodies recognizing PRMT1 or PRMT5 and against phospho-serine/threonine, an approach previously used to assess changes in PRMT1 phosphorylation[36]. As shown in Fig. 6f, positive PLA puncta were found

in control cells for both PRMT1 and PRMT5, which were significantly reduced upon *ULK3* silencing. As a complementary approach, we took advantage of a chromatography resin that preferentially binds phos-phoproteins and is used for microscale phosphoprotein purification[63]. Expression vectors for tagged PRMT1 or PRMT5 were transfected alone or in combination with one for ULK3, and the cellular lysates were bound to the resin. The eluted fractions were separated by SDS-PAGE and analyzed by western blotting using FLAG antibodies. Indicative of enhanced phosphorylation, the amounts of PRMT1 and PRMT5 in the resin-bound fractions were significantly increased in cells with ULK3 overexpression (Fig. 6g).

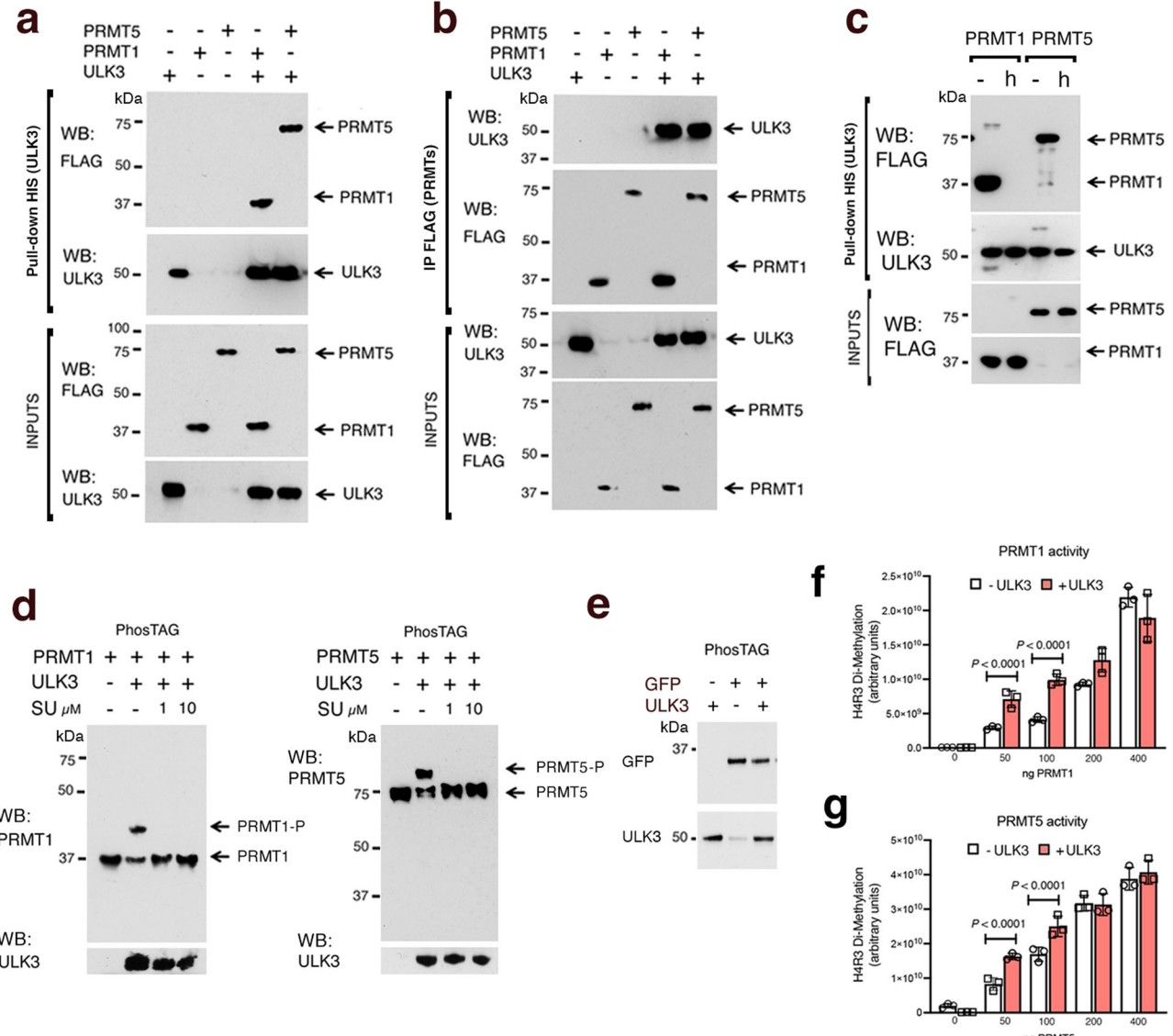

**Fig. 7 | ULK3 binds directly to PRMT1 and PRMT5 and controls their activity.**
**a** Immunoblot analysis of recombinant FLAG-tagged -PRMT1 (200 ng) or -PRMT5 (200 ng) admixed with His-tagged ULK3 (200 ng) and pulled-down with a Ni-NTA resin (NTA). The same amounts of proteins were incubated with the resin singularly as controls. Sequential FLAG and ULK3 immunoblots. **b** Recombinant PRMT1 and PRMT5 were incubated with ULK3 as above before PRMT pull-down with anti-FLAG antibodies. Immunoblots were probed as indicated. **c** Recombinant PRMT1 and PRMT5 plus/minus heating for 20 min at 90 °C (**h**) before incubation with ULK3, NTA pull-down, and sequential immunoblot analysis as in (**a**). **d** Recombinant PRMT1 and PRMT5 were incubated with recombinant ULK3 for 30 minutes at 37°C in kinase buffer containing 200 μM ATP. Proteins were separated on a Phos-TAG gel, which delays the migration of phosphorylated proteins[64], followed by immunoblot analysis with the indicated antibodies. Parallel kinase reactions contained 1 and 10 μM ULK3 inhibitor SU6668[43]. The immunoblot analysis of the same samples in Fig. 7d separated by SDS-PAGE is in Supplementary Fig. 7a. A kinase assay with

PRMT1 plus/minus heat treatment prior to ULK3 kinase assay is in Supplementary Fig. 7b. **e** In vitro ULK3 kinase assay using recombinant GFP as substrate. Proteins were admixed as in (d) before western blot analysis using the indicated antibodies. **f, g** Impact of ULK3 on PRTM1 and PRMT5 activity. Increasing amounts of PRMT1 or PRMT5 (0, 50, 100, 200, 400 ng) were incubated with (red bars) or without ULK3 (white bars) as in (d), followed by the methylation reaction (with 50μM S-adenosyl-methionine) of recombinant histone 4 (H4) (0.5 μg). Quantification of H4 arginine 3 asymmetric-, for PRMT1 activity (**f**), or symmetric-, for PRMT5 activity (**g**), dimethylation by dot blots analysis with antibodies specific for these epigenetic modifications. Mean ± SD, $n$(dot blot/condition) = 3, $P < 0.001$, two-tailed unpaired $t$-test. Supplementary Fig. 7c shows symmetric and asymmetric dimethylated arginine immunoblottings of methylation assays using ULK3 as a substrate. Supplementary Fig. 7d shows an immunoblot with the same antibodies of ULK3 and truncated ULK3 IP from transfected HEK293 cells.

Thus, ULK3 associates with the PRMT1 and PRMT5 arginine methyltransferases and can determine their phosphorylation state.

## ULK3 directly binds PRMT1 and PRMT5 and increases their methylase activity

An attractive possibility is that ULK3 can interact and phosphorylate PRMT1/5 directly. To assess this possibility, purified recombinant His-tagged ULK3 was admixed with FLAG-tagged PRMT1 or PRMT5

proteins, followed by immunoprecipitation with the respective anti tag antibodies. Both PRMT1 and PRMT5 were effectively recovered after the pulldown of ULK3 and the latter was recovered after the PRMT1 and PRMT5 pulldown (Fig. 7a, b). Notably, the ULK3 - PRMT1/5 interaction required a native protein conformation as it was lost when the PRMTs were heat denatured before performing the binding assay (Fig. 7c).

To assess whether PRMT1/5 can be a direct substrate of ULK3 activity, we incubated the recombinant proteins in the presence of ATP

followed by Phos-TAG SDS-polyacrylamide gel electrophoresis, which detects phosphorylated proteins as retarded band(s)[64]. A large fraction of PRMT1 and PRMT5 was found in slower migrating forms upon incubation with ULK3, which were not detected when the in vitro incubation assay was performed in the presence of the ULK3 inhibitor SU6668[43] (Fig. 7d and Supplementary Fig. 7a). The gel retardation form was abolished when PRMT1 was heat-denatured before the kinase reaction, pointing to the importance of a native protein conformation (Supplementary Fig. 7b). No gel delayed forms were detected in parallel assays with recombinant GFP protein, indicating the specificity of the reaction (Fig. 7e).

An important question was whether ULK3 could directly affect the activity of PRMT1 and/or PRMT5. Accordingly, we preincubated ULK3 with increasing amounts of PRMT1 or PRMT5 proteins, followed by the analysis of their enzymatic activity towards histone H4. Arginine 3 of histone 4 (H4R3) is a substrate for both PRMT1-driven asymmetric (H4R3DA) and PRMT5-driven symmetric (H4R3DS) dimethylation activity[32]. Dot blot analysis using specific anti -H4R3DS and -H4R3DA antibodies showed a significant increase of the methylated products at the lower concentrations of the two enzymes when pre-incubated with ULK3, indicative of increased activity (Fig. 7f, g). Immunoblot analysis of ULK3 immunoprecipitated after in vitro preincubation assays with antibodies specific for asymmetric and symmetric dimethylated arginine showed that the kinase is in turn methylated by the PRMT1 but not PRMT5 enzyme (Supplementary Fig. 7c), with a selectivity of modification that was also observed after co-expression assays in HEK293 cells (Supplementary Fig. 7d), with functional implications that will have to be separately pursued.

Thus, ULK3 can bind and phosphorylate PRMT1 and PRMT5 directly, affecting their activity.

## ULK3 is required for PRMT1 and PRMT5 recruitment to chromatin and H4 arginine 3 dimethylation

PRMT1[36] and PRMT5[65] are key for the self-renewal of HKCs and SCC cells[66,67]. As assessed by immunoblotting, total protein levels of PRMT1 and PRMT5 in SCC13 cells were not significantly affected by *ULK3* gene silencing (Fig. 8a). Cell fractionation followed by immunoblotting confirmed that the ULK3 protein is mainly nuclear in SCC13 cells and showed that the nuclear localization of PRMT1 and PRMT5 was not affected by *ULK3* silencing (Fig. 8b), with results confirmed by immunofluorescence analysis (Fig. 8c).

To assess whether the association of PRMTs with chromatin is affected by ULK3 loss, SCC13 cells plus/minus *ULK3* silencing were processed for ChIP with anti -PRMT1 and -PRMT5 antibodies, followed by qPCR amplification of different regions of the *PKM* and *LDHA* metabolic genes, which are bound by ULK3. We found two main regions of PRMT1 and PRMT5 binding to the *LDHA* and *PKM* genes, which significantly reduced in cells with silenced *ULK3* (Fig. 8d).

To assess whether the results reflected a more generalized impact of *ULK3* loss, total chromatin was immunoprecipitated with anti-histone H3 antibodies followed by immunoblotting with antibodies against PRMT1, PRMT5, and histone H3. As shown in Fig. 8e the association of both PRMTs with chromatin was strongly reduced in *ULK3*-silenced cells versus controls.

Histone H4 arginine 3 (H4R3) is dimethylated by PRMT1 and PRMT5[32]. The decreased chromatin association of PRMTs resulting from *ULK3* gene silencing was associated with the downregulation of both symmetric and asymmetric forms of H4R3 dimethylation, as assessed by western blotting and immunofluorescence analysis with corresponding specific antibodies of SCC and HeLa cells with silenced *ULK3* versus control. (Fig. 8f, g and Supplementary Fig. 8a–d). Consistent with the knockdown results, *ULK3* loss by CRISPR/Cas9 deletion resulted in a similar decrease in both forms of H4R3 dimethylation (Fig. 8h and Supplementary Fig. 8e).

Thus, ULK3 is required for the association to chromatin of PRMT1/5 and for dimethylation of their histone H4 substrate.

## ULK3 is a target of translational significance to suppress SCC cells proliferation and oncogenic potential

An important question was whether targeting ULK3 could effectively suppress SCC formation. In the first set of studies, we utilized orthotopic mouse ear injection assays of fluorescently labeled cancer cells, which allow in vivo imaging of lesions[18,68]. Two different SCC cell lines, the mildly transformed SCC13 cell line[69] and the more aggressive SCCO28 cell line[70], were stably transduced with a GFP-expressing lentivirus, followed by infection with two *ULK3*-silencing lentiviruses versus empty vector control virus before ear injections. While control SCC cells formed rapidly expanding tumors, cells in which *ULK3* was downregulated gave rise to tumors of smaller size and greatly reduced number (Fig. 9a, b).

As an alternative approach, we employed an intradermal injection assay of SCC cells in Matrigel, which allows the formation of easily retrievable localized nodules[68]. Even in this case, tumors formed by SCC cells with *ULK3* silencing were significantly smaller than those derived from control cells, with tumor growth being suppressed entirely in some animals (Fig. 9c). The decreased tumorigenicity was mirrored by a reduction in the Ki67 proliferative index and increased differentiation, as indicated by the large increase of cancer cells with the KRT10 marker expression (Fig. 9d, e). Consistent with the findings with cultured cells, tumors formed by SCC13 cells with silenced *ULK3* had significantly reduced levels of TP63 expression and H4R3 dimethylation relative to control lesions (Fig. 9f, g).

An exciting tool with translational potential to suppress the expression of target genes is provided by locked nucleic acids (LNAs), oligonucleotide analogs of high stability amenable to in vivo delivery[71,72]. SCC13 cells were transfected with three LNAs designed to target different regions of the ULK3 mRNAs (#1–3) in parallel with scrambled controls (SCR). RT-PCR and immunoblot analysis showed effective downmodulation of *ULK3* expression by all three LNAs, with only one of them (#1) also causing a slight reduction of the other ULK family members (Fig. 10a, b). Consistent with the gene silencing and CRISPR-deletion results, the colony formation of SCC cells was significantly decreased by administration of these LNAs (Fig. 10c), in parallel with a significant reduction of their sphere-forming capability, another well-established assay of cancer stem cell potential (Fig. 10d).

ULK3 has been reported to delay cytokinesis of cells under stress conditions at the G2/M phase of the cell cycle, by phosphorylating a component of the ESCRT-III cytosolic complex[17]. We assessed the impact of ULK3 loss on cell cycle distribution by analyzing SCC cells expressing a dual color Fluorescence Ubiquitin Cell Cycle Indicator system (FUCCI)[73]. Transfection of these cells with the anti-ULK3 LNAs led to an accumulation of cells in the G1/S phases of the cell cycle (expressing the RFP reporter), with little or no effect on G2/M (expressing the GFP reporter) (Fig. 10e). At the same time, ULK3 silencing did not result in detectable nuclear alterations and/or multi-nucleated cells as would be expected in case of a mitotic block (Fig. 10f).

LNA administration also recapitulated the in vivo consequences of stable *ULK3* gene silencing. SCC13 cells transfected with two ULK3-targeting LNAs (#2 or #3) 24 h before intradermal injection into mice formed much smaller tumors than controls (Fig. 10g) with a reduction of Ki67 proliferative index and TP63-positive cells and enhancement of differentiation (KRT10-positive areas) (Fig. 10h, i).

Therefore, targeting ULK3 is of translational value to suppress the oncogenicity of SCC cells impacting the balance between growth and differentiation.

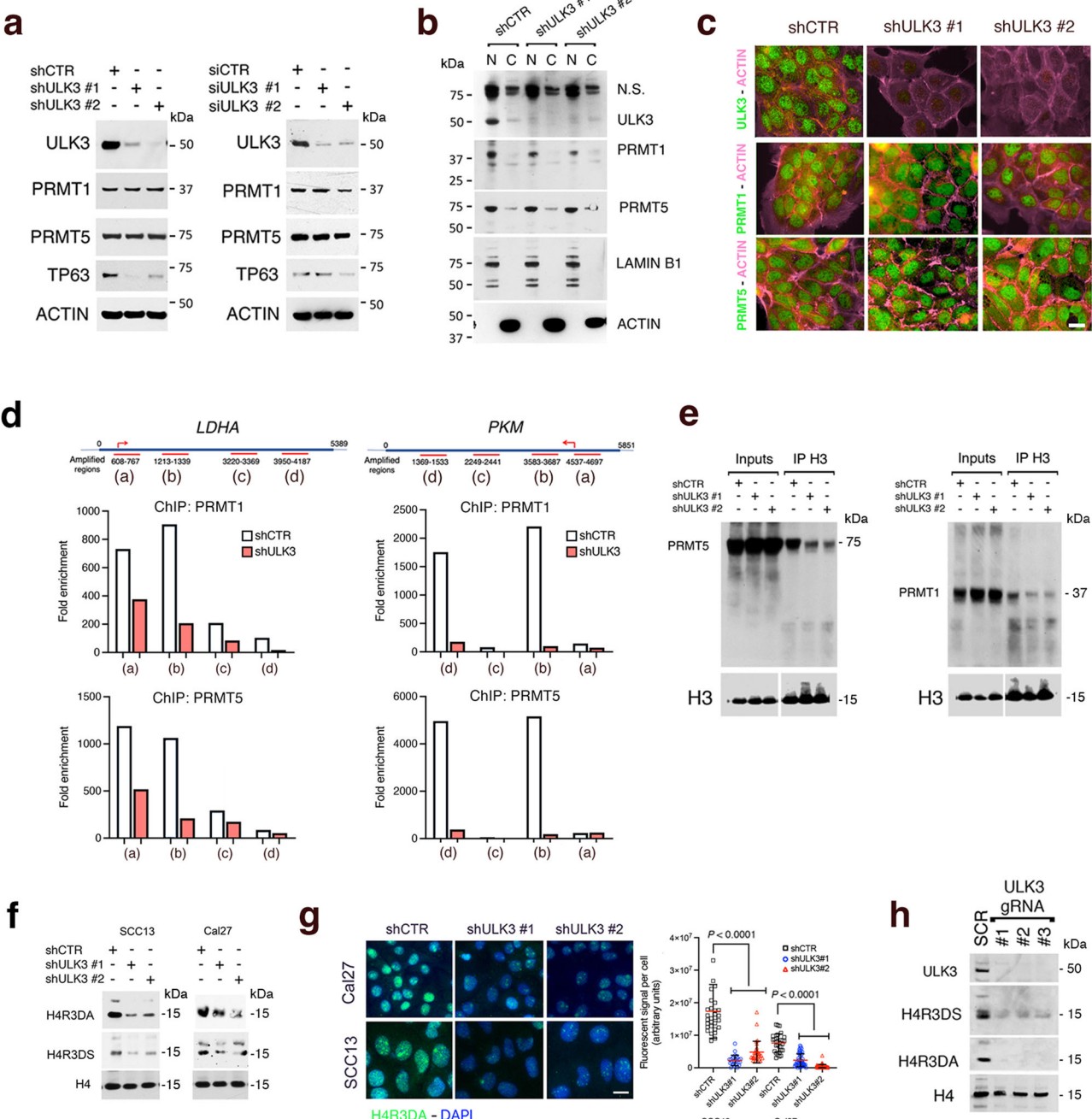

**Fig. 8 | ULK3 loss impairs PRMT1/5 chromatin association and function.**
**a** Immunoblots against the indicated proteins of SCC13 cells with *ULK3* silencing by two shRNAs (shULK3 #1, #2) for 1 week or with two siRNAs (siULK3#1, 2) for 48 h, with corresponding controls. **b** Immunoblots of nuclear (N) and cytosolic (C) fractions from SCC13 cells with *ULK3* silencing by two shRNA-specific versus control lentiviruses. The blots were sequentially probed with the indicated antibodies. **c** PRMT1, PRMT5 and ULK3 immunostaining (green) of SCC13 cells plus/minus *ULK3* silencing with two lentiviruses versus control virus. Phalloidin labeled actin cytoskeleton (magenta). Scale bar 5 μm. **d** ChIP with anti -PRMT1 and -PRMT5 antibodies from SCC13 cells plus/minus *ULK3* silencing followed by qPCR amplification of four regions of the *PKM* and *LDHA* genes. The position of amplified regions (magenta lines) and TSS (arrow) are shown. Results are expressed as enrichment fold relative to IgGs as in[18]. **e** ChIPs with anti-histone 3 (H3) antibodies followed by PRMT1, PRMT5 and H3 immunoblotting from SCC13 cells plus/minus *ULK3* silencing. For H3, the immunoblots of inputs and immunoprecipitations are shown at different

exposure times. **f** Asymmetrically (H4R3DA) and symmetrically (H4R3DS) dimethylated arginine 3 of histone 4 immunoblotting in SCC cells plus/minus *ULK3* silencing for one week, with anti-histone H4 as a loading control. An additional H4R3DS and H4R3DA immunoblot upon *ULK3* silencing is in Supplementary Fig. 8a. **g** H4R3DA (green) immunofluorescence of two SCC cell lines plus/minus *ULK3* silencing, with DAPI (blue) as nuclear staining. Representative images and quantification of individual cells fluorescence signal, Cal27: *n*(cell/condition) = 35; SCC13: *n*(cells/condition) = 30, mean ± SD, *P* < 0.0001, one-way ANOVA. Scale bar 5 μm. An additional experiment is in Supplementary Fig. 8b. Supplementary Fig. 8c, d show immunoblot and immunostaining with anti -H4R3DS, -H4R3DA, and H4 antibodies of HeLa cells plus/minus ULK3 silencing. **h** H4R3DA, H4R3DS, total H4 and ULK3 immunoblotting of SCC13 cells with CRISPR/Cas9-mediated *ULK3* gene deletion with three guide RNA (gRNA) versus control (SCR), as in Fig. 2d. A similar analysis of HEK293T cells plus/minus CRISPR/Cas9-mediated *ULK3* gene deletion is in Supplementary Fig. 8e.

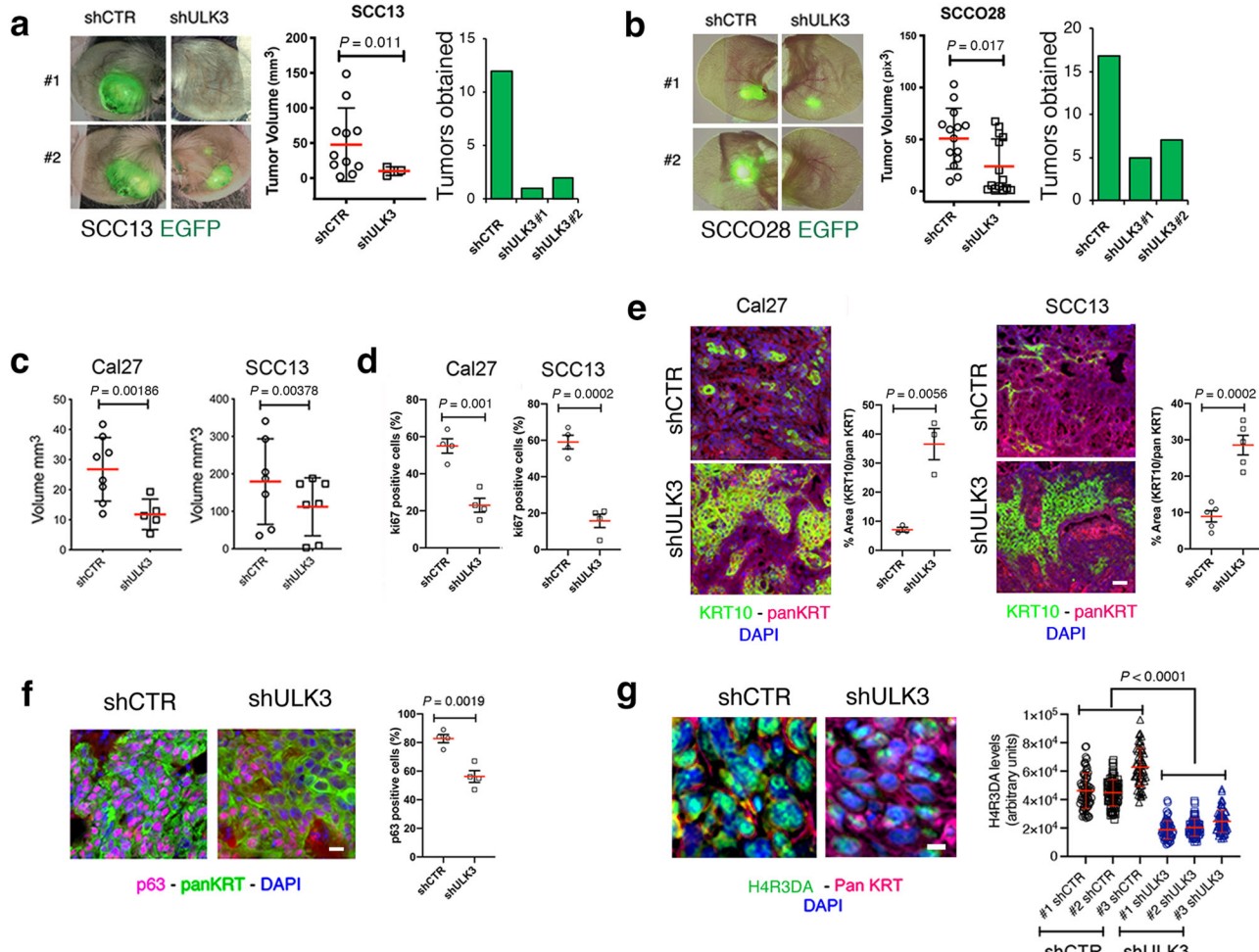

**Fig. 9 | ULK3 downregulation suppresses SCC growth in vivo. a, b** Ear injection skin SCC model. SCC13 (**a**) and SCCO28 (**b**) cells expressing an enhanced green fluorescence protein (EGFP) were infected with two *ULK3*-silencing or control lentiviruses. Mice (NOD/SCID-Prkdc[scid]; CB-17) received contralateral ear injections of either *ULK3*- or control- silenced cells. Representative fluorescence with phase-contrast image superimposition of two ear pairs three weeks after injection and quantification of all tumor volumes (V = (length × width$^2$) × 0.5)) for SCC13 cells (**a**). For SCCO28 cells (**b**) the quantification of all tumor volumes was as (integrated density × area of GFP fluorescence, normalized to Day 1). $n$(mice/SCC cell line) = 20, mean ± SEM, $P < 0.01$, two-tailed unpaired *t*-test. Green histograms show the number of tumors obtained with each lentivirus after three weeks. **c-g** Intradermal backskin injection SCC model. Cal27 and SCC13 cells plus/minus *ULK3*-silencing were injected (8/condition) with Matrigel in contralateral back sites of mice (NOD/SCID IL2Rγ$^{-/-}$). **c** Tumor volumes were measured as in (a). Mean ± SEM, SCC13 $n$(tumors/shCTR) = 7, $n$(tumors/shULK3) = 7; Cal27 $n$(tumors/shCTR) = 8, $n$(tumors/shULK3) = 5, $P < 0.001$, two-tailed unpaired t-test. **d** Quantification of Ki67 immunofluorescence of tumors formed by SCC cells plus/minus *ULK3* silencing. Anti-pankeratin (panKRT) antibodies identified injected cells and DAPI stained

the nuclei. Mean ± SEM, three fields per tumor, $n$(tumors/condition) = 4, $P < 0.001$, two-tailed unpaired t-test. Scale bar 30 μm. **e** Keratin 10 (KRT10, green) and panKRT (magenta) immunofluorescence of the SCC tumors as above. Representative images and quantification of fluorescence signal (area (%)/field), mean ± SEM, three fields per tumor, SCC13: $n$(tumors/condition) = 5, Cal27: $n$(tumors/condition) = 3, $P < 0.001$, two-tailed unpaired t-test. Scale bar 30 μm. **f** TP63 (magenta) and panKRT (green) immunofluorescence of tumors formed by SCC13 cells plus/minus *ULK3* silencing. Representative images and quantification of fluorescence signal (% TP63 + panKRT positive cells/section), mean ± SEM, four fields per tumor, $n$(tumors/condition) = 3, $P < 0.001$, two-tailed unpaired t-test. Scale bar 30 μm. **g** H4R3DA (green) and panKRT (magenta) immunofluorescence, with DAPI nuclear staining, of SCC13 cell tumors plus/minus *ULK3* silencing. Representative images and quantification of single cell nuclear fluorescent signal, shCTR: $n$(cells/ tumor #1) = 66, $n$(cells/ tumor #2) = 72; shULK3#1: $n$(cells/ tumor #1) = 72, $n$(cells/ tumor #2) = 58, $n$(cells/ tumor #3) = 66; shULK3#2: $n$(cells/ tumor)#1 = 68, $n$(cells/ tumor #2) = 58, $n$(cells/ tumor #3) = 67, mean ± SEM, $P < 0.001$, one-way ANOVA. Scale bar 5 μm.

## Discussion

The proliferation and differentiation of squamous epithelial cells are tightly regulated processes, with genetic and epigenetic alterations in regulatory networks often leading to a transformed phenotype[2,3]. We show here that ULK3, a kinase we previously reported to be key for the activation of cancer-associated fibroblasts[18], plays an equally important role in the epithelial compartment, as a transcription regulator of self-renewal and oncogenic potential. In skin SCCs, ULK3 is pre-eminently expressed in the nuclear compartment, overlapping with epidermal areas of sustained cell proliferation and TP63 expression. Pronounced nuclear accumulation of ULK3 was also found in actinic

keratosis lesions, a pathological condition of major clinical significance that can evolve into SCC[3,74], pointing to a role for this kinase in the early events of the keratinocyte transformation process. Large clinical data sets analysis showed that *ULK3* expression levels are also elevated in lung, head/neck, and cervical SCCs. The findings are likely of functional significance, as ULK3 targeting decreased HKC and SCC proliferative potential, promoting differentiation and limiting tumor growth.

Our findings establish ULK3 as a key regulator of the transcription process underlying the balance between growth and differentiation of both HKCs and SCC cells with a concomitant impact on metabolism. *ULK3* gene silencing induced the cell cycle inhibitor *CDKN1A*, a

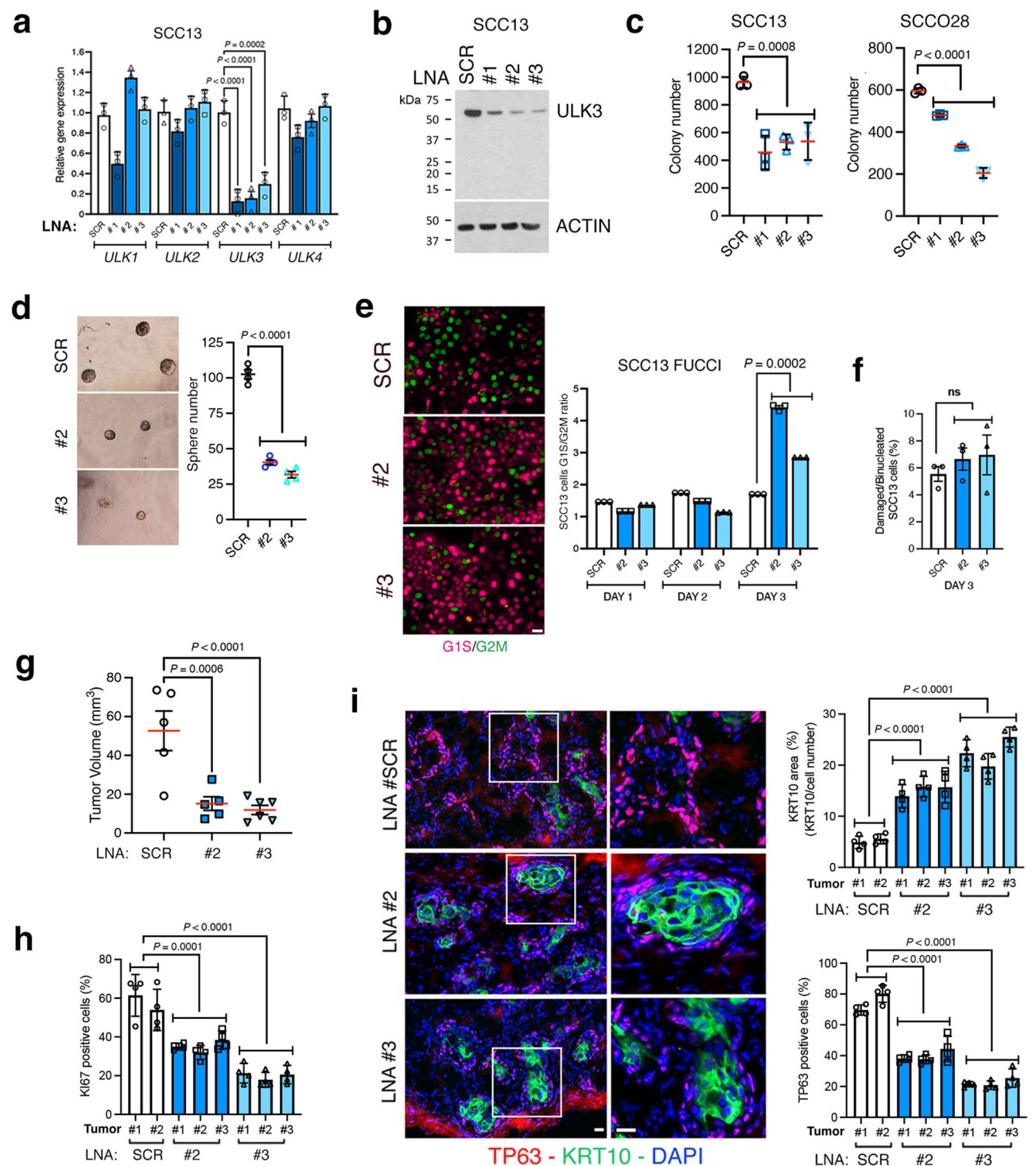

mediator of differentiation-associated cell cycle withdrawal[44], and several key markers of keratinocyte terminal differentiation process, *KRT1, KRT10, IVL,* and *FLG*[6,46,75]. Gene set enrichment analysis (GSEA) of transcriptional profiles of multiple HKC strains plus/minus *ULK3* silencing showed a strong negative correlation between *ULK3* expression and gene signatures involved in KC proliferation and oncogenesis, such as increased ΔNp63, Hh/GLI and MYC signaling, as well as gene signatures related to SCC aggressiveness. The transcription factor TP63 plays a key role in the transition from simple to stratified epithelia and in maintaining skin stem cell populations[22]. Like *TP63, GLI2* encodes for a transcription factor with a positive role in the oncogenic conversion of keratinocytes for both BCCs and SCCs[12,76].

ULK3 has a central role in GLI1/2 activation by SHH or TGFβ in human adipose stem cells[43] and the elevated ULK3 levels we found in SCCs might be the reason for GLI activation in SCCs.

Downregulation of these key programs can explain the negative consequences of loss of ULK3 on self-renewal potential of HKCs and SCCs, as assessed by the well-established clonogenicity and sphere-forming assays[77]. Consistent results were obtained by shRNA-mediated *ULK3* gene silencing, CRISPR-deletion, and treatment with a pharmacological inhibitor of ULK3 activity. Abrogation of ULK3 expression by LNAs, a complementary approach of translational significance[72], exerted a similar impact on SCC proliferative potential, with an accumulation of cells in the G1 phase of the cell cycle, and suppressing

**Fig. 10 | ULK3 targeting with locked nucleic acid (LNAs) suppresses SCC self-renewal and tumorigenicity. a** RT-qPCR analysis of ULK family members (*ULK1-4*) in SCC13 cells 72 h after transfection with three locked antisense nucleic acids (LNAs) targeting *ULK3* (#1-3) or scrambled control (SCR). Mean ± SEM, $n$(experiments) = 3, $P < 0.001$, $P < 0.0001$, two-tailed unpaired $t$-test. **b** ULK3 and ß-ACTIN immunoblots of SCC13 cells treated as in (**a**). **c** Colony forming assays of SCC cell lines transfected as in (**a**) and plated at low-density for one week, $n$(dishes) = 3, mean ± SEM, $P < 0.0001$, one-way ANOVA. **d** Sphere-forming capability of SCC13 cells transfected with ULK3-targeting LNAs (#2, #3) or control (SCR), grown one week on Matrigel. Representative phase-contrast images and quantification of sphere number, $n$(dishes) = 4, mean ± SEM, $P < 0.0001$, one-way ANOVA. **e** Cell cycle distribution of SCC13 cells expressing fluorescence ubiquitin cell cycle indicators (FUCCI)[73] after transfection with ULK3-targeting LNAs (#2, #3) or scrambled control (SCR). Day 3 representative images and quantification of G1/S (magenta) versus G2/M (green) reporter ratio over time (days). Mean ± SEM, $n$(dishes) = 3. Day 1: $n$(cells SCR) = 1405: $n$(cells LNA#2) = 1099, $n$(cells LNA#3) = 1551; Day 2: $n$(cells SCR) = 2205, $n$(cells LNA#2) = 2667, $n$(cells LNA#3) = 1266; Day3: $n$(cells SCR) =

1215, $n$(cells LNA#2) = 940, $n$(cells LNA#3) = 1726, $P < 0.0001$, one-way ANOVA. Scale bar 5 μm. **f** Percentage of cells treated as in (**e**) with nuclear damages (cytosolic DNA) or multinucleated, as detected by DAPI staining. Mean ± SEM, $n$(dishes/condition) = 3, $n$(cells SCR) = 860, $n$(cells LNA#2) = 906, $n$(cells LNA#3) = 911, ns: not significant, one-way ANOVA. **g** SCC13 cells 24 h after transfection with ULK3-targeting LNAs (#2, #3) or scrambled control were injected with Matrigel intradermally into the back of NOD/SCID (IL2Rγ$^{-/-}$) mice for 10 days. Tumor size was calculated as (V = (length × width$^2$) × 0.5)). Mean ± SEM, $n$(tumors/SCR) = 5, $n$(tumors/LNA) = 6, $P < 0.001$, and $P < 0.0001$, two-tailed unpaired $t$-test. **h** Ki67 and panKRT (to identify SCC cells) immunofluorescence of tumors described in (**g**), and DAPI staining. Percentage of Ki67 positive SCC13 cells, 4 fields/tumor, $n$(tumors/SCR) = 2, $n$(tumors/LNA) = 3, mean ± SEM, $P < 0.01$, $P < 0.001$, one-way ANOVA. **i** TP63 (magenta) and KRT10 (green) immunostaining of tumors described in (**g**). Low and high magnification representative images and quantification of TP63-positive cells and KRT10 area (%), normalized to the number of cells, 4 fields/tumor, $n$(tumors/SCR) = 2, $n$(tumors/LNA) = 3, mean ± SEM, $P < 0.0001$, one-way ANOVA. Scale bar 30 μm.

---

tumorigenesis, with a parallel suppression of TP63 expression and induction of differentiation.

Treatment of SCC cells with mTOR or PI3K inhibitors - as already tested in clinical trials - causes suppression of glycolysis and a concomitant increase of glutamine utilization as an alternative source of energy[29]. A similar compensatory mechanism occurs in SCC cells upon *ULK3* gene silencing, which results in the suppression of genes with a key role in glycolysis (*PKM, PDK2/3, TIGAR,* and *LDHA*) and the upregulation of genes involved in glutamine uptake and utilization (*GLS1* and *SLC1A5*). This gene regulation was consistent with metabolomics analysis showing, upon *ULK3* silencing, an enrichment of several amino acid metabolic pathways, including glutamate, glutathione, and glycine/serine metabolism. Thus, as with inhibitors already considered for SCC treatment[29], targeting ULK3 activity/levels may benefit from a combinatory use of glutaminolysis inhibitors.

While most previous studies focused on ULK3 cytosolic function[14,16–18], we have found that ULK3 in HKCs and SCC cells is mostly localized to nuclei and, by combined ChIP-seq and ChIP assays, binds to a specific set of genes of functional significance, including several with a key role in metabolism and transcription. Notably, most of the ULK3-bound and/or regulated genes exhibited a striking loss of H3K27 acetylation upon *ULK3* silencing pointing to its role in the maintenance of open chromatin configuration. Other chromatin regulatory mechanisms are likely to be also affected by ULK3 silencing, as indicated by the fact that H3K27 acetylation is not compromised by the ULK3 loss at all ULK3-bound genes.

Considering the impact on the larger number of genes modulated upon ULK3 silencing compared with the genes directly bound, it poised us to identify additional mechanisms by which ULK3 impinges on transcription. The recruitment of epigenetic modifiers to chromatin regulates chromatin activation states (activated or repressed)[20,78]. We identified the histone arginine methylases PRMT 1 and 5 (PRMT1/5), which are the most represented in HKCs and contribute to transformation, proliferation, and survival in numerous cancers[79]. PRMT1 is predominantly a transcriptional coactivator[32,80], while PRMT5 frequently acts as a corepressor[32,81]. However, opposite functions have been reported for PRMT1[82] and PRMT5[83]. Thus, PRMT1/5 function is likely directed by their interacting partners and is, to some extent, cell-type specific[36,78].

Both PRMTs converge to regulate transcription as part of a methylome complex[33] and a functional redundancy exists among PRMT1/5, as PRMT1 loss or pharmacologic inhibition causes substrate "scavenging" by PRMT5[84,85]. In HKCs, PRMT1 interacts with CSNK1A to sustain cell proliferation and repress premature differentiation[36] while PRMT5 suppresses PKCδ/p38-induced involucrin expression[60].

By multiple approaches, we showed that PRMT1 and PRMT5 directly associate with the Ct-terminal region of ULK3, that such

binding requires PRMTs native conformation and is independent of active ULK3 kinase activity since it also occurs with kinase-defective point mutants. Functionally, we demonstrated that the enzymatic activity of PRMT1/5 is enhanced upon phosphorylation in vitro by ULK3. The findings with our GSEA which revealed that the loss of *ULK3* in HKCs phenocopied the changes in gene expression induced by *PRMT1* silencing, further supported the functional convergence.

The exact consequences of mutually exclusive PRMT5-driven symmetric or PRMT1-driven asymmetric arginine dimethylation of the same arginine residues are poorly understood[32]. Both PRMTs function in the epigenetic modification of the arginine 3 on histone 4 (H4R3), which can impact the recruitment of several writers, including EP300, to the chromatin[34,35,86]. *ULK3* downregulation, or gene deletion, blunted such epigenetic modifications in skin and oral SCCs, HeLa cells, and HEK293 cells, suggesting a specific and conserved mechanism. Mechanistically, we found that ULK3 loss decreases the recruitment of PRMT1/5 to the promoter areas of specific genes and, more globally, to chromatin.

It should be noted that so far, neither ULK3 nor PRMT1/5 have been identified in major large-scale functional genomic screening assays for fitness or dependency genes in SCC cells. Although these screens are instrumental in identifying targets, they critically depend on the depth and efficiency of the targeting gene libraries and, as a consequence, other well-known genes involved in the control of SCC development, such as NOTCH1 and CDK6, have often been missed[87,88].

Overall, the present work identifies a control mechanism involving the serine/threonine kinase ULK3 and the PRMT chromatin complex, which impinges on key transcriptional programs and the metabolism of keratinocyte and SCC cells. LNAs can effectively downmodulate ULK3 expression and exert similar effects on self-renewal and tumorigenicity of SCC cells as *ULK3* gene silencing. This kinase represents an attractive target for the prevention and treatment of SCC lesions that would allow the simultaneous suppression of cancer cells and adjacent cancer-associated fibroblasts[89,90].

## Methods

### Mouse, cells, and human samples
All the animal procedures were performed as approved by the Institutional Animal Care and Use Committee of Massachusetts General Hospital (MGH), Boston, MA, USA (MGH # 2004N000170).

Normal primary human keratinocytes (HKCs) were prepared from discarded human samples from abdominoplasty at the Cutaneous Biology Research Center (MGH, Boston, MA, USA), as approved by Institutional Review Board protocol (IRB # 2018P003156) or were previously obtained with approval in refs. [46,68]. All HKC strains were identified by a progressive number or two letters identifying the operators. Normal human keratinocytes and patient-derived

squamous cell carcinoma (SCC) cells were cultured as previously described in refs. [46,68].

Human skin samples of actinic keratoses (AK), skin SCCs and normal matched controls of nearby non-affected areas from Mohs surgeries of skin cancers (excess of areas accumulating at the end of the incision after the wound closure) were obtained at the Department of Dermatology (MGH, Boston, MA, USA) as de-identified discarded parts not needed for diagnosis as approved by the IRB protocol (# 2018P003156). A progressive number identified all samples. Both male and female skins were used, as available.

Dr. Rocco, (MGH, Boston, Massachusetts, USA) provided oral SCCO28 cells. Dr. Tolstonog (Centre Hospitalo-Universitaire Vaudois, Lausanne, Switzerland) provided Cal27, Cal33, and FaDu oral SCC cells[68]. Dr. Rheinwald (Brigham and Women's Hospital, Boston, Massachusetts, USA) provided skin SCC cells (SCC12 and SCC13). All cells were routinely tested for the absence of mycoplasma.

### Tissue arrays
Commercially available tissue arrays with normal or adjacent matched tissues as controls of skin squamous cell carcinoma (SK483), head and neck tumors (HN801c) and cervical squamous carcinomas (CR484) were from US Biomax. SK483 contains skin squamous cell carcinoma tissues with normal tissues as control, 48 cases-48 cores. HN801c contains head and neck squamous cell carcinoma tissues, 40 cases-80 cores. CR484 contains cervix squamous cell carcinoma tissues with matched adjacent cervix tissue, 16 cases-48 cores. Slides were processed with routine histology procedures as in refs. [18,68] using rabbit anti ULK3 ATLAS antibody (AB_2677003, 1:100 dilution).

For signal quantification, acquired images for each color channel were imported into ImageJ software and converted into binary images. The intensity was measured using "measurement" in areas of interest. The signal was normalized to the number of cells, as detected by using a fluorescent DNA stain (4′,6-diamidino-2-phenylindole) to label nuclei. Images were prepared using Adobe Photoshop 2021, and statistical analysis was made using Prism 9 software.

### Cell manipulations and assays
Conditions for culturing HKCs, SCCs, and other cell lines, RT-qPCR analysis, lentivirus production in 293 T cells and infection of cells were as in refs. [46,62,68].

For the apoptosis assays, SCC13 and Cal27 cells infected with two separate lentiviruses for *ULK3* shRNA or a control virus were seeded $2 \times 10^3$/3 cm dish in triplicate and, after two days, incubated for 1 h with apoptosis/necrosis detection kit reagents (Abcam, cat # Ab176749). The apoptosis-detecting sensor (PS) has green fluorescence (Ex/Em = 490/525 nm) upon binding to the membrane. Loss of plasma membrane integrity after necrosis allows a membrane-impermeable 7-AAD (Ex/Em = 546/647 nm) dye to label the nucleus. Live cells were labeled with CytoCalcein Violet 450 (Ex/Em = 405/450 nm). Images of live cell cultures were obtained using an inverted fluorescent microscope Leica and quantified using ImageJ.

For clonogenicity assays, HKC, HeLa, or the indicated SCC cells (infected either with two separate *ULK3* or a control shRNA-expressing virus and selected with puromycin for three days) were plated ($10^3$/ 3 cm dish) in triplicate or quadruplicate/condition and cultured for six days before staining the colonies with 1% crystal violet. Images of the stained dishes were quantified using ImageJ.

For proliferation assays, we seeded an equal number of cells ($10^3$/ 3 cm dish) in triplicate. We assessed the cell number of HKCs and SCC cells (plus/minus infection with shULK3- or shCTR- virus and selection for three days) on days 3 and 5 using a hemocytometer.

For CRISPR/Cas9-mediated gene deletion, all-in-one lentiviral vectors encoding both the RNA guide (gRNA) and the Cas9 protein for ULK3 gene deletion were from ABM GOOD. We infected SCC13 and HEK293 cells with three separate gRNA (#1, #2, #3) (or the combinations indicated) or a scrambled control guide vector (#SCR). After three days, we selected the cells for three additional days with 1 μg puromycin (Gibco) and then pooled the colonies before the analysis of ULK3 disruption, colony assays and western blot analysis.

The Surveyor Assay[41] of SCC13 cells was made one week after the infection with lentiviruses co-expressing the Cas9 enzyme and one of the three RNA guides targeting *ULK3* (gRNA ULK3 #1-3) or a scrambled control gRNA (SCR gRNA). We amplified by PCR the genomic regions of each targeted sequence (using separate oligo pairs #1-3) and tested 1 μg of amplified genomic DNA for the presence of a DNA-mismatch by incubating with a single-strand DNAse using a commercial kit Thermo Fisher (#A24372). The reactions of each targeted sequence were analyzed with a 2% agarose gel in parallel with a similar reaction made with SCR gRNA-infected cells, as a specificity control.

For ULK3 overexpression assays, Cal27 and SCC13 cells were infected with a lentiviral wild-type ULK3-HA expressing vector (ABM Good), or an empty vector, overnight then selected with 1 μg/ml of puromycin (Gibco) for three days. After the expansion of the cultures for one week $10^3$/3 cm dish cells were plated in triplicate or quadruplicate cultures and analyzed for colony assay formation after a week. Crystal violet-stained images of cell cultures were analyzed by ImageJ and Prism software for statistical evaluation.

For the spheroid assay, SCC13 and Cal27 cells infected with a lentiviral GFP-ULK3 wt vector, or an empty control vector (Genecopeia) were plated cells ($10^3$/well) onto 8-well chamber slides coated with 100 μl Matrigel (BD Biosciences) in normal culture medium plus 2% Matrigel. After one week, the colonies were quantified using ImageJ. Cultures of control and ULK3-overexpressing SCC cells were seeded in parallel dishes, and the gene expression was analyzed after two days by RT-qPCR.

For locked antisense RNA (ASO LNA) transfection (IDT), we incubated overnight SCC13, SCC13 FUCCI, Cal27 or SCCO28 cells with complexes formed by mixing each LNA (10 nM final concentration) with 10 μl lipofectamine 2000 (Invitrogen) for 30 min before adding to the cell medium. We used three LNAs targeting ULK3 open reading frame and scrambled control (SCR). The following day cells were trypsinized, counted, and used for biochemical studies, sphere assays or in vivo tumor growth.

For spheroid assays of SCC13 cells, $10^2$ cells 24 h after transfection with LNA or control oligo were seeded in quadruplicate in an 8-well culture slide coated with Matrigel and allowed to grow for seven days. We quantified the number of spheres obtained with ImageJ.

SCC13 FUCCI is a cell line stably expressing fluorescent ubiquitination-based cell cycle indicator (FUCCI) reporter, which allows the detection of G1/S (magenta fluorescence) and G2/M (green fluorescence) phases of the cell cycle[73]. For each time point (24 h, 48 h, and 72 h) of experiment $10^3$ cells/3 cm dishes were seeded in triplicate. Cells were fixed with 4% paraformaldehyde in PBS and stained with DAPI to evaluate the total cell number. The fluorescence signals were acquired using a digital slide scanner (Nanozoomer S60, Hamamatsu) and quantified by ImageJ.

To determine the requirement of ULK3 kinase activity, the clonogenicity and cell proliferation assays were similarly carried out by adding to the culture medium, the day after the seeding, 10 μM of SU6668 (5-[1,2-Dihydro-2-oxo-3H-indol-3-ylidene)methyl]−2,4-dimethyl-1H-pyrrole-3-propanoic acid) an inhibitor of ULK3 kinase activity[19] (Biotechne cat # 3335), or DMSO carrier (Sigma) as a negative control.

*The shRNA sequences used are provided in* Supplementary TABLE 1. *The sequences of the oligonucleotides for ChIP and qPCR are in* Supplementary TABLE 2. *The sequences of siRNA and LNAs are in* Supplementary TABLE 3. *The Vectors and Recombinant proteins are in* Supplementary TABLE 4. *The detailed list of antibodies used is in* Supplementary TABLE 5.

## Immune detection

Western blots and immunofluorescence analysis were performed as in[46,62]. Immunohistochemistry (IHC) of tumors and tissue sections was performed as in[18,46] and the images were acquired using a Leica or a Zeiss confocal microscope. For immunofluorescence, IHC and western blot analysis we used anti-ULK3 from Santa Cruz (# 137897, 1:100 dilution) and Abcam (# EPR4888, 1:100 dilution), anti-symmetrical dimethylation of arginine 3 on histone 4 from EpiGentek (# 10019-4R3DS,1:100 dilution), anti-asymmetrical dimethylation of arginine 3 on histone 4 from EpiGentek (# 10019-4R3DA, 1:100 dilution), anti-TP63 from Abcam (# 735, 1:100 dilution), anti-Ki67 from Abcam (# 15580, 1:100 dilution), anti-KRT10 from Covance (# 19054, 1:100 dilution). For western blot and immunofluorescence analysis the antibodies for PRMT1 and PRMT5 were from Cell Signaling (# 2449, 1:1000 dilution) and (# 79998, 1:1000 dilution) respectively. For western blot analysis we used Cell Signaling anti -GLS1 (# 88964, 1:1000 dilution), -PKM1 (# 7067, 1:1000 dilution) and -p21 (# 2947, 1:1000 dilution), and Abcam anti-INVOLUCRIN (#227530, 1:1000 dilution) antibodies. For Immunofluorescence analysis anti-VIMENTIN antibodies (Abcam # 20346, IF 1:100 dilution).

For the nuclear/cytosol separation, total lysates from SCC13 cells stably infected with two different shRNA targeting ULK3, or a control virus, were fractionated using the NE-PER Nuclear and Cytoplasmic Extraction Kit (Thermo Scientific #78833). As determined by a separate stained Coomassie gel, equal amounts of nuclear and cytosolic protein extracts were blotted and the membranes sequentially incubated with antibodies against ULK3, PRMT1, PRMT5, LAMIN B1 and ß-ACTIN (all 1:1000 dilution).

## Protein interactions and immunoprecipitations

Proximity ligation assays[61] for detecting endogenous interactions were performed using a Duo-link PLA kit (Sigma, # DUO92101) according to the manufacturer's protocol and as in refs. [18,62]. After incubation with PLA blocking solution, the SCC cells were incubated with primary anti -ULK3 (MyBioSource # MBS9200567, 1:100 dilution) and -PRMT1 (Cell Signaling # 2449, 1:100 dilution) or -PRMT5 (Cell Signaling # 79998, 1:100 dilution) antibodies. The cells were incubated with the PLA probes, anti-rabbit PLUS, anti-mouse MINUS, washed, ligated, and amplified by rolling circle amplification. Images were obtained with a Nikon Eclipse Ti confocal microscope.

PLA analysis of PRMT1 and PRMT5 phosphorylation in HeLa cells plus/minus ULK3 silencing was similarly carried out by using anti-phospho Ser/Thr antibodies (Abcam #17464, 1:100 dilution) in combination with anti -PRMT1 Santa Cruz (# 166963, 1:100 dilution), -PRMT5 (Santa Cruz # 424245, 1:100 dilution), or nonimmune control antibodies (1:100 dilution), with the addition of phosphatase inhibitors (Thermo Fisher) in all the buffers.

We performed the immunoprecipitations of endogenous or transiently transfected proteins in SCC, HeLa, and HEK293 cells with anti -ULK3 (Santa Cruz # 517373, 1:100 dilution), -PRMT1(Cell Signaling # 2449, 1:100 dilution), -PRMT5 (Cell Signaling # 79998, 1:100 dilution), -FLAG (Sigma #F104, 1:100 dilution) and -V5 (Thermo Fisher E10, 1:100 dilution) antibodies using the same conditions as reported in refs. [18,62].

For ChIP followed by western blot experiments, SCC13 cells infected with two ULK3 silencing lentivirus, or a control virus, were crosslinked with 1% formaldehyde and processed for ChIP using the conditions indicated in SimpleChiP assay kit (Cell Signaling # 56383, 1:100 dilution). The crosslinked chromatin from $16 \times 10^6$ cells per condition was sonicated and equal amounts of total chromatin (50 µg), as assessed by spectroscopy analysis of purified DNA, were immunoprecipitated using anti-histone H3 (Cell Signaling # 4620, 1:100 dilution) or nonimmune control (Cell Signaling, 1:100 dilution) antibodies. After the crosslink reversal for 2 h at 56°C the immunocomplexes were analyzed by western with anti -PRMT1, -PRMT5 and -H3 antibodies (1:1000 dilution).

The in vitro protein interactions using the baculovirus-expressed proteins were performed as follows. FLAG-tagged PRMT1 (100 ng) (Active Motif #31411) or PRMT5 (100 ng) (Active Motif #31393) were mixed with His-tagged ULK3 (100 ng) (Abcam #101548) in binding buffer (150 mM NaCl, 100 mM Tris ph8, 0.5% NP40, 10% Glycerol) for 4 h at 4°C on a rotating platform in the presence of 20 µl protein G magnetic beads, anti FLAG (Sigma #F104, 1:100 dilution) and Dyna-beads (Thermo Fisher) or HisPur Ni-NTA magnetic beads (Thermo Fisher). After this time, the complexes were washed three times in binding buffer w/o glycerol, once in binding buffer high salts (300 mM NaCl) and resuspended in 20 µl of SDS-PAGE loading buffer for electrophoresis. We carried out the western blots using anti -ULK3, -FLAG, -PRTM1 and -PRMT5 antibodies (1:1000 dilution). To determine the requirement of a native configuration in some experiments the recombinant PRMTs were heated for 20 min in a thermal block, while the same amount of PRMTs protein was left on ice as control, before mixing with the ULK3 protein.

For the kinase assays 1 µg of recombinant ULK3 was mixed with 1 µg of recombinant PRMT1 or PRMT5 in kinase buffer (25 mM Tris ph 7.5, 5 mM ß-glycerophosphate, 2 mM DTT, 0.1 µM NaVO4, 10 mM MgCl2) containing 200 µM ATP and incubated for 30 min at 30°C. The reaction was stopped by adding 1/2 volume of SDS PAGE loading buffer. Parallel reactions were carried out in the absence of ULK3 kinase as negative, untreated controls. Some kinase assays were performed in the presence of 1 or 10 µM SU6668 ULK3 kinase inhibitor (Biotechne cat # 3335), or DMSO carrier (Sigma) as a negative control. All the samples from the kinase assays were separated on 12.5% SuperSep™ pre-cast gel copolymerized with SuperSep™ Phos-Tag™ (50 µmol/l), 7.5% acrylamide (Wako-FUJI, #192–18001). After the separation, gels were washed for 30 min in 10 mM EDTA in transfer buffer before the blotting. The western blots were carried out using anti -ULK3, -GFP, -FLAG, -PRTM1 and -PRMT5 antibodies.

In vitro Methylation assays were performed by adding 500 ng of ULK3 to 500 ng of PRMT1 or PRMT5 in methylation buffer (50mMTris pH 8.6, 0.02% Triton X100, 2 mM MgCl2, 1 mM TCEP) containing 50 µM SAM for 3 h at room temperature. Reactions were stopped with 1/2 reaction volume of SDS PAGE loading buffer. The western blots were carried out by using antibodies recognizing di-methylated symmetric (Cell Signaling #13222, 1:1000 dilution) or asymmetric arginine (Cell Signaling #13522, 1:1000 dilution), anti -FLAG and -ULK3 (1:1000 dilution).

The PRMTs activity was measured as follows: PRMT1 (1 µg) or PRMT5 (1 µg) were incubated with ULK3 (1 µg) (or without ULK3) in kinase buffer for 30 min at 30 C. After this time, 0, 50, 100, 200, 400 ng of PRMT1 or PRMT5 were diluted in methylation buffer containing 500 ng/assay of recombinant histone H4 (Active Motif #31493). Reactions were performed in 25 µl and stopped after 3 h by adding 5 µl of SDS PAGE sample buffer 6x. For each reaction, equal volumes (3 µl) were spotted on a nitrocellulose membrane in triplicate and incubated overnight with antibody anti H4R3 symmetric (EpiGentek# 10019-4R3DS, 1:1000 dilution) or H4R3 asymmetric (EpiGentek# 10019-4R3DA, 1:1000 dilution) arginine di methylation. The dot blots were developed with ECL reagents (Pierce), the autoradiographic films quantified using ImageJ, and the values (in arbitrary units) normalized to the highest enzymatic activity obtained in the presence of ULK3.

## Tumorigenesis experiments

**Ear injections.** *Mouse-ear* injections with the SCC13 and SCCO28 SCC cells were carried out in 8 to 10-week-old female NOD/SCID mice (*CB17sc-m*, Taconic), as in[18]. Both cell lines were infected with a high-titer EGFP-expressing lentivirus[18] and then with either two different ULK3 silencing shRNAs or a control vector as in[18]. A total of $10^5$ ULK3- or control-silenced EGFP-expressing SCC cells were injected 5 µl per site using a 33-gauge micro syringe (Hamilton) in the contralateral ear of the same animal. Two experiments with 10 mice each were injected

for the SCC13 cells and the SCC028 cells. The day after injection all the mice ears were imaged using a fluorescent stereomicroscope (Leica MZ-FLIII) to confirm the efficient injection and the images were used to normalize the fluorescence intensity at the end of the experiment (21 days). At that time images of the ears were taken using a bright field and fluorescence stereomicroscope. For SCC13 cells tumor sizes were measured using a digital caliper using the following formula: $V = (length \times width^2) \times 0.5$, and for SCCO28 cells the volume was calculated by determining the fluorescence intensity as in[18] using ImageJ software.

**Back injections.** Mouse back intradermal tumorigenicity injection assays were carried out in 10-week-old NOD/SCID female mice (with IL-2 receptor γ-chain null mutation, *NOD.Cg-Prkdcscid Il2rytm1Wjl/SzJ*, The Jackson Laboratory), as in ref. [68]. Assays were conducted with skin SCC13 or oral Cal27 SCC cells infected with ULK3 silencing shRNA #2 or with a control shRNA virus, selected and amplified in 1 μg/ml of puromycin. In each injection site we introduced 100 μl of HBSS containing $10^6$ cells diluted 50% with reduced growth factor Matrigel (BD biosciences, cat # 356234) using an insulin syringe. Cells, reagents and syringes were kept on ice before the injection. Each animal was injected in two sites on the same side of the back skin per combination. Three weeks after injection the tumors were removed, imaged, measured, and further processed for immunochemistry.

Intradermal injection with LNA-treated SCC13 was carried out as follows. SCC13 cells were transfected separately with two ULK3 targeting LNAs (#2 and #3) or with scrambled control (#SCR) (10 nM final concentration). After 24 h, 8-week-old NOD/SCID female mice (IL-2 receptor γ-chain null mutation, *NOD.Cg-Prkdcscid Il2rytm1Wjl/SzJ*, The Jackson Laboratory) were injected with 100 μl of HBSS containing $10^6$ cells diluted 50% with reduced growth factor Matrigel (BD biosciences, cat # 356234) using an insulin syringe. Animals were injected in two sites of the back skin per combination. Ten days after the injection the tumors were removed, imaged, measured using a digital caliper using the following formula: $V = (length \times width^2) \times 0.5$ and frozen in optimal cutting temperature compound (OCT, Scigen Tissue-Plus, Thermofisher) before processing for immunochemistry.

The maximal tumor size permitted by the approved IACUC protocol was 20 mm in any direction, which was not exceeded in any experiment. All mice were euthanized by carbon dioxide inhalation. No animal died or was excluded from the experiments. Mice were housed in four per cage on a 12 h dark/light cycle, with a constant ambient temperature of 65–70 F and 40% humidity. All the mice were randomly allocated to the experimental groups and group-housed.

**Transcriptomic, Clariom D assays and GSEA**
Transcriptomic profiles from Head and Neck (H&N), Lung, and Cervical SCCs from the TCGA Research Network updated to 2018 were plotted as Violin plots generated using individual sample values (z scores, calculated according to $z = (expression\ in\ tumor\ sample\ -\ mean\ expression\ in\ reference\ sample) / standard\ deviation\ of\ expression\ in\ reference\ sample)$. Head and neck: $n(normal) = 44$, $n(SCC) = 526$; lung SCCs: $n(normal) = 45$, $n(SCC) = 509$; cervical SCCs: $n(normal) = 13$, $n(SCC) = 374$.

TCGA Research Network datasets updated to 2016 as in[2] of primary tumors versus normal tissues from Bladder, Cervical, Esophageal and Lung SCCs. The analysis of these datasets was made using GEPIA software (http://gepia.cancer-pku.cn/index.html) and shows box-and-whiskers plots, with indicated the median, box (25–75%) and whiskers (5–95%), log(transcript per million+1), value cutoff $P < 0.01$.

BLCA: $n(normal) = 19$, $n(SCC) = 404$; CESC: $n(normal) = 3$, $n(SCC) = 306$; ESCA: $n(normal) = 13$, $n(SCC) = 182$; LUSC: $n(normal) = 50$, $n(SCC) = 484$.

Transcriptomic studies were performed using Clariom D arrays (Applied Biosystem). Total RNA was extracted by using Directzol RNA

Miniprep assay Kit (Zymo Research #R2071) one week after ULK3 shRNA silencing for HKCs (three separate HKC strains silenced each with two ULK3-targeting shRNAs, or a control plasmid) and two weeks after ULK3 silencing for SCC13 cells (silenced with two separate ULK3-targeting shRNAs, or a control vector). Library probe preparation was done with 1 μg of total RNA. Clariom D arrays were used on the GeneChip™ 3000 instrument system and the results were processed with Transcriptome Analysis Console (TAC) software to analyze and visualize global expression patterns of genes, pathways, and possible alternative splicing events.

Gene set enrichment analysis (GSEA) for Clariom D array expression profiles was performed by using GSAA-SeqSP software (gene set association analysis for array expression data with sample permutation) from the Gene Set Association Analysis (GSAA) platform (version GSAA_2.0, http://gsaa.unc.edu/). Curated gene sets were obtained from the Molecular Signatures Database (MSigDB v5.2; http://www.broadinstitute.org/gsea/msigdb/).

DAVID software (https://david.ncifcrf.gov/) was used for gene ontology (GO) analysis on lists of genes modulated by ULK3 silencing in HKCs and SCC13 cells.

**Metabolomics studies**
Analysis of steady-state metabolites in SCC13 cells infected with two separate lentiviruses for ULK3 shRNA, or a silencing control virus, was performed at Beth Israel mass spectrometry core facility. In triplicate, one 10 cm culture dish per condition was extracted three times with 500 μl 80% cold methanol (−80°C), (each extraction was made on ice for 10 min, followed by centrifugation and resuspension of the pellet in 80% methanol, the three extractions pooled before next step) then the extracted metabolites were centrifuged and dried using a speed vacuum concentrator for 30 min with heat. Dried pelleted samples were resuspended in 100 μl of water before analysis at the Beth Israel Deaconess Mass Spectrometry Core facility directed by Dr. J. Asara.

Metabolites were evaluated individually for their significance or by Metabolite Set Enrichment Analysis to identify biological patterns significantly enriched using MetaboAnalist software (http://www.metaboanalyst.ca/).

**ChIP-seq and ChIP analysis**
For ChIP-seq, chromatin prepared from $32 \times 10^6$ SCC13 cells as in ref. [18] by using a simple ChIP assay kit (Cell Signaling # 56383), was immunoprecipitated using 10 μg anti ULK3 (Santa Cruz # 517373, 1:100 dilution) or nonimmune control antibodies (Santa Cruz, 1:100 dilution), followed by quantification of the DNA by fluorometry on a Qubit system (Invitrogen) and quality/size determination using the Bioanalyzer assay (Agilent). Chromatin from SCC13 plus/minus ULK3 silencing was similarly immunoprecipitated using 5 μg of anti-H3K27ac antibodies (Abcam # 4729, 1:100 dilution), or non-immune control antibodies (Abcam, 1:100 dilution) and analyzed as above.

According to the manufacturer's recommendations, a total of 10 ng DNA was used for library preparation using the NEBNext ChIP-Seq Library Prep Kit (Illumina).

Raw data files from ChIP-seq assays were aligned to the GRCh38 genome (https://www.ncbi.nlm.nih.gov/assembly/GCF_000001405.26/) with Bowtie2 Version 2.3.0 (http://bowtie-bio.sourceforge.net/bowtie2). Duplicates were removed with Picard (https://broadinstitute.github.io/picard/) and, for peak detection, MACS2 software (http://liulab.dfci.harvard.edu/MACS) was used with a p-value cutoff of 1.00e − 04. Peaks were annotated with HOMER (http://homer.ucsd.edu/homer/index.html). The Integrative Genomics Viewer (http://software.broadinstitute.org/software/igv/) was used to illustrate ChIP-seq peaks.

DAVID software (https://david.ncifcrf.gov/) was used for gene ontology (GO) analysis.

Data sets generated for this study were deposited in the NCBI Gene Expression Omnibus.

**ChIP experiments.** SCC13 cells were infected with two viruses targeting ULK3 (shULK3#1 and shULK3#2) or a control virus (shCTR), selected and amplified in 1 μg/ml of puromycin. Cell cultures were crosslinked with 1% formaldehyde, and total cellular chromatin was obtained using ChIP sonication kit solutions (Cell signaling# 56383). The efficiency of the sonication process was monitored by using 1% agarose gels. For validation of ULK3 binding to promoters and of active chromatin regulation by ULK3 silencing, 30 μg of crosslinked sonicated chromatin was immunoprecipitated overnight with anti ULK3 (Santa Cruz # 517373, 1:100 dilution), H3K9ac (Upstate #06-942, 1:100 dilution), H3K27ac (Abcam # 4729, 1:100 dilution), Pol II (Upstate #05-623, 1:100 dilution) or nonimmune antibodies (Cell Signaling, 1:100 dilution), as control, in the presence of 40 μl of magnetic beads protein G (Cell Signaling). After three washes in low salts and one wash in high salt in nuclear lysis chromatin preparation buffers (Cell Signaling), the chromatin-bound to antibodies was eluted for 30 min at 65 °C and de-crosslinked for 2 h at 65 °C in the presence of 2 μl Proteinase K (10 mg/ml). PCR was carried out by using for each analyzed gene (*PKM, LDHA, FOXM1*) two separate oligonucleotide pairs amplifying the regions of the promoter/TSS.

To determine the consequences of ULK3 silencing on PRMT1 and PRMT5 association to specific promoters, 50 μg of crosslinked chromatin prepared as described above from SCC13 cells plus/minus ULK3 silencing with one shRNA (shULK3 #2), were immunoprecipitated overnight with 5 μg of rabbit anti PRMT1 (Cell Signaling # 2449), anti PRMT5 (Cell Signaling # 79998) or nonimmune IgGs (Cell Signaling). For each ULK3-bound metabolic gene analyzed, (*PKM* and *LDHA*), the immunoprecipitated chromatin was amplified with oligonucleotide pairs designed to amplify four different regions within the ULK3 binding peak, as determined by ChIP seq analysis. ChIP experiments were carried out in parallel duplicates. Fold enrichment was calculated by normalizing to IgG values using the $2^{-\Delta Ct}$ formula.

### Statistics and reproducibility
Data are presented as mean ± SEM, mean ± SD, or ratios among treated and controls, with three or more separate HKC strains and SCC cells, or patient-derived samples, used in independent experiments, as indicated in the Figure legends. All western blots were performed at least twice as independent biological replicates. When a representative image is shown, the number of samples and conditions analyzed are provided in the figure legend. For gene expression and all functional testing assays, the statistical significance of differences between experimental groups and controls was assessed by two-tailed unpaired or paired *t*-test, one-way ANOVA, and by comparing to random permutations (for GSEA), as indicated in the legends. In all cases, *P* values < 0.05 were considered statistically significant. Statistical significance analysis and plotting of the data were performed using Microsoft Excel 365 v16.67 and Prism 9 Graphpad software.

### Reporting summary
Further information on research design is available in the Nature Portfolio Reporting Summary linked to this article.

## Data availability
The transcriptomic (HKCs and SCC13 cells) and ChIP-seq data generated in this study have been deposited in GEO under the accession codes GSE183084, GSE183085, and GSE183933, respectively. The metabolomics data have been deposited in MetaboLights as MTBLS5155. Sequences for shRNA, siRNA, LNAs, CRISPR/cas9, PCR oligos, and antibodies used are in Supplementary Tables 1–5. Source data underlying Figs. 1a–h, 2a–f, 3a–j, 4b–g, 5c–f, 6b–g, 7a–g, 8a, b, d–h, 9a–g, 10a-i and Supplementary Figs. S1d, S2a, S2d, S2e, S2f, S4, S5d, S6e, S7a–c, S8s, S8d are provided with this paper. Source data are provided with this paper.

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

## Acknowledgements

We are grateful to Dania Al Labban and Esther Revai for help with the experiments, and to Enzo Calautti, Caterina Missero for critical reading of the manuscript. The head and neck, lung, bladder, esophageal and cervical data shown in Fig. 1a are based upon data generated by the TCGA Research Network: www.cancer.gov/tcga. This work was supported by grants from the National Institutes of Health (R01AR078374; R01AR039190; content not necessarily representing the official views of NIH). GPD is a member of the SKINTEGRITY.CH collaborative research program and students in his group are supported by funding from the European Union's Horizon 2020 research and innovation program under the Marie Skłodowska-Curie grant agreement No 859860.

Emery Di Cicco is a recipient of a "Fellowship for Abroad 2020" from Fondazione AIRC.

## Author contributions

S.G., A.C., G.B., B.T., E.D and M.M. performed experiments and analyzed the results with G.P.D. A.C., B.T. and S.G. performed the bioinformatic analysis of transcriptomic with G.P.D. S.G., A.C., M.M., C.S. and Novogene, UK, conducted the bioinformatics analysis of Chip-seq experiments with G.P.D. S.D. and V.N. provided the clinical samples. S.G. and G.P.D. designed the study and wrote the manuscript.

## Competing interests

The authors declare no competing interests.
