## [Peer Review File · Nature Communications]

The ULK3 kinase is a determinant of keratinocyte self-renewal and tumorigenesis targeting the arginine methylomeREVIEWER COMMENTS

Reviewer #1 (Remarks to the Author):

Goruppi and colleagues report on a proposed role for ULK3 in squamous cell carcinoma. On the face of it the data look convincing in support of a role for ULK3 in squamous cell carcinoma survival but there are numerous issues with claims made about self-renewal, differentiation and tumorigenesis.

The authors quote some of the prior ULK3 literature but seem to ignore aspects of this work that could certainly explain some of the observed effects after genetic manipulation of ULK3. Furthermore, authors findings are surprising since this particularly signaling axis has not been identified in multiple studies of squamous cell carcinoma, a good example of which is Campbell et al, Cell Reports, 2018 23(1):194-212.e6. doi: 10.1016/j.celrep.2018.03.063. PMID: 29617660, who identify multiple signaling nodes that include TP63 but do not include ULK3 or PRMT1/5. While this reviewer appreciates different approaches to analysis will yield different results it is surprising that studies have not previously highlighted ULK3. Review of the Human Protein Atlas identifies ULK3 with low tissue specificity, low overall expression levels in multiple tissues, including cancer of the head and neck, skin and other SCC prone tissues and a favorable prognostic marker in pancreatic cancer (ULK3 protein expression summary - The Human Protein Atlas, <https://www.proteinatlas.org/ENSG00000140474-ULK3>). Again, different approaches to analysis yield different results but in order to determine whether the analysis presented in Figure 1 is accurate it would be helpful to understand how z-scores were calculated from TCGA data.

Since it has been reported that ULK3 regulates cytokinesis (reference 19) a more plausible explanation for the described phenomena might be that down regulating/ deleting ULK3 inhibits mitosis. As such the paper describes that inhibiting a regulator of mitosis inhibits cell growth, which explains a possible role in tumorigenesis. It would seem more proliferative tissues express more ULK3 which, given its role in mitosis, is somewhat expected.

Authors seem to imply that ULK3 is a regulator of cell fate but the data supporting this are weak and are most likely a result of regulating mitosis regardless of self renewal/ differentiation in the context of stem cell fate. Inhibiting keratinocyte proliferation usually results in differentiation since this is the keratinocytes default pathway.

Use of ULK3 inhibitors (Kasak et al., Characterization of Protein Kinase ULK3 Regulation by Phosphorylation and Inhibition by Small Molecule SU6668. Biochemistry. 2018 Sep 18;57(37):5456-5465. doi: 10.1021/acs.biochem.8b00356. PMID: 30096229) would be of interest in the context of the study.

Statements regarding binding of ULK3 to PRMT1 and 5 are over-interpreting the data (summary) since PLA, Mass spec and IP all identify association/interaction but not direct binding. More detail on the results of the Mass-spec experiment would be helpful; how many peptides identified, how do PRMT1 and 5 sit amongst other proteins identified by this unbiased approach?

All generated datasets should be deposited in public databases, no access was given to GSE183933 for reviewers so it is not clear which generated datasets have been deposited.

Referencing is rather author centric and key molecular studies on SCC are lacking.

Reviewer #2 (Remarks to the Author):

Goruppi et al. introduce ULK3 as a master regulator for SCC cell proliferation and tumorigenesis. The arginine methyltransferases PRMT1 and PRMT5 are identified as ULK3 interactors and substrates. Furthermore, the impact of PRMT activity on chromatin configuration and translation is analysed. The authors conclude with suggesting ULK3 and the PRMTs as pharmacologic targets for the treatment of SCCs.

While the findings are novel and convincingly presented, several aspects in the long and detailed line of evidence might need additional experimental support.

Comments:

Line 130 The authors describe a novel signalling pathway with ULK3 regulating proliferation, colony formation and tumorigenesis. The mechanism is in place both in SCC cells and HeLa cells. It is worth discussing why this fundamental role of ULK3 as a fitness gene has not been detected in the numerous genome-wide siRNA or CRISPR screens.

Line 138 The arrangement of panels in Fig 2 can be made more consistent. Same color code in the graphs - explain why the respective cell lines have been chosen - either show the western blots or omit. Maybe combine panels d to j?

Line 154 Please discuss whether or not this result is expected when set in relation to the ULK3 literature.

Line 163 Please relate the relative expression from the WB quantification (Fig 2d-j, Fig 3e-f) with the relative expression from the MS quantification. Maybe present the MS analysis first - then the WB analysis?

Line 229 Please compare the identified ULK3 interaction partners to previously published ULK3 interactors.

Line 236 ULK3 has been introduced as a nuclear protein (Fig 1c). The ULK3:PRMT1/5 speckles, however, are distributed evenly in nucleus and cytoplasm. In addition to showing the PLA co-localization in Fig 5b, it will be good to also show the subcellular distributions of ULK3 and PRMT1/5.

Line 247 While ULK3 K44H represents a common inactivating point mutation, ULK3 K139R is likely to be catalytically active. Please discuss / explain.

Line 252 The microscopic PLA is too indirect to establish PRMT1/5 as cellular ULK3 substrates. Additional MS analysis of the enriched phosphoproteins is suggested.

Line 401 Please discuss the translational significance in more detail since this is also prominently mentioned in the abstract. Are ULK3 or PRMT1/5 inhibitors available? Will it be advantageous to apply PROTACs over inhibitors? What are potential adverse effects when targeting these proteins pharmacologically?

And there were some typos (lines 142 159 202 360 Fig 2g Fig 2j)...

Reviewer #3 (Remarks to the Author):

Sandro et. al demonstrated that ULK3 kinase is upregulated in SCCs, and knockdown of ULK3 attenuates the expression of genes related to oncogenic potential and glycolysis. The oncogenic role of ULK3 was validated in orthotopic models of skin cancer. Mechanistically, ULK3 was found to be associated with PRMT1 and PRMT5, and to be required for their function on chromatin. There are quite a few issues need to be addressed.

1. Abstract is muddled. It is not clear how H4R3 regulates chromatin configuration and cell differentiation as they are two forms of H4R3 methylation, symmetrical and asymmetrical, with distinct functions. I suggest a re-write.

2. In addition to Fig. 2D, genomic DNA sequencing was also needed to confirm the knockout of ULK3 in cells.

3. The author mentioned that Δ Np63 regulates both proliferation and self-renewal in Fig. 2F. However, the previous experiments only touched the growth aspect.

4. It might be better to move Fig. 2F-2J to Fig. 3 as panel A.

5. The change of Δ Np63, gli1/2, CDKN1A, KRT1, KRT10, IVL, and FLG at protein level needs to be shown.
6. Line 148-149, "Thus, ULK3 is an essential determinant of HKC and SCC cell proliferative potential that affects other critical regulators of the growth/differentiation of these cells". I don't see enough evidence for a link between ULK3 and differentiation.
7. Transcriptomic analysis needs to be done with two independent shRNAs to ensure that the target genes identified for ULK3 are reliable.
8. Modulation of metabolism-related genes upon ULK3 silencing should be confirmed by RT-qPCR and western blot analysis in HKC cell and SCC cells.
9. Steady-state metabolomic analysis need to be done to test the effects of ULK3 in SCC028 cells.
10. In addition to steady-state metabolomic analysis, the levels of metabolites, such as glutamate and glutathione, should also be examined.
11. In addition to H3K27ac, more histone markers are required to support the effects of ULK3 knockdown on chromatin openness.
12. Why those 57 genes showed no sign of reduced H3K27ac level when ULK3 was knocked down?
13. What's the change of H3K27ac in TIGAR, PDK2, PDK3, SLC2A1, SLC1A5, SLC38A6 and GLS1 promoter region? Does ULK3 bind to these genes from ChIP-seq analysis?
14. ChIP-seq analysis of metabolic genes need to be validated by RT-qPCR.
15. How is the overlap of genes regulated by PRMT1/5 and ULK3? What are the enriched GO terms for genes co-regulated by PRMT1/5 and ULK3?
16. In Fig. 5C, PRMT1 seems to bind to ULK3 with higher affinity than PRMT5 when ULK3 was served as the bait. However, when PRMT1 and PRMT5 were served as baits, ULK3 seems to bind equally well with PRMT1 and PRMT5.
17. In Fig5d, it seems that PRMT1 binds to ULK3K139R with higher affinity than WT and K144H, while PRMT5 binds to K144H with the highest affinity. Please clarify.
18. The author proposed that PRMTs might be substrates for ULK3, but why not the other way around, namely ULK3 serves as a substrate for PRMTs? Rescue experiments might be helpful to determine the relationship between PRMTs and ULK3.
19. In Fig.5F, it is unclear how could those antibodies used can specifically detect the phosphorylated PRMT1?
20. The overlap between PRMT- and ULK3-bound genes needs to be shown.
21. Which regions in PRMT1 and PRMT5 are phosphorylated by ULK3?
22. In Fig 6E, author found that knockdown of ULK3 reduced the global levels of H4R3 dimethylation. Does ULK3 affect PRMT1 methyltransferase activity?
23. Whether knockdown of ULK3 affects PRMT1 and 5 binding to those metabolic genes?
24. Whether mutation of the phosphorylation sites in PRMT1 and 5 could affects their chromatin binding needs to be tested?
25. Effects of PRMT1 and PRMT5 on metabolism should be tested by steady-state metabolomic analysis.
26. Whether PRMT1 and PRMT5 downregulation affects SCC growth in vivo needs to be tested?

REVIEWER COMMENTS

Reviewer #1 (Remarks to the Author):

Goruppi and colleagues report on a proposed role for ULK3 in squamous cell carcinoma. On the face of it the data look convincing in support of a role for ULK3 in squamous cell carcinoma survival but there are numerous issues with claims made about self-renewal, differentiation and tumorigenesis. The authors quote some of the prior ULK3 literature but seem to ignore aspects of this work that could certainly explain some of the observed effects after genetic manipulation of ULK3.

Reply: We thank the reviewer for finding our paper of interest and for the recommendations on how to improve it, which we have addressed as specified below, also considering the possibility that the ULK3 involvement in squamous differentiation is mediated by some other previously reported roles.

Authors findings are surprising since this particularly signaling axis has not been identified in multiple studies of squamous cell carcinoma, a good example of which is Campbell et al, Cell Reports, 2018 23(1):194-212.e6. doi: 10.1016/j.celrep.2018.03.063. PMID: 29617660, who identify multiple signaling nodes that include TP63 but do not include ULK3 or PRMT1/5. While this reviewer appreciates different approaches to analysis will yield different results it is surprising that studies have not previously highlighted ULK3. ...Again, different approaches to analysis yield different results but in order to determine whether the analysis presented in Figure 1 is accurate it would be helpful to understand how z-scores were calculated from TCGA data.

Reply: We thank the reviewer for pointing out the importance of considering previous studies, in particular the work by Campbell et al., 2018, which we have considered in detail. A likely explanation why ULK3 was not included as a potential key node by Campbell et al., is their decision to focus only on genes consistently upregulated across all the cancer types that were analyzed. We have looked at the raw data and found that ULK3 levels are elevated in 4 of the 5 cancer types that were analyzed, including cervical, esophageal and lung SCCs as well as bladder cancer. We have added an additional panel to Figure 1 panel a (left) to illustrate these findings, with gene expression data expressed as transcripts per million as in Campbell's paper. The right panel, based on TCGA data, was retained as such, specifying in the figure legend that z-scores are calculated according to the formula: $z = (\text{expression in tumor sample} - \text{mean expression in reference sample}) / \text{standard deviation of expression in reference sample}$, with an indication of the TCGA website address where more information about their operational definition can be found (<https://gdc.cancer.gov/> : "For mRNA and microRNA expression data, we typically compute the relative expression of an individual gene and tumor to the gene's expression distribution in a reference population. That reference population is either all tumors that are diploid for the gene in question, or, when available, normal adjacent tissue.").

In addition, as we point out in the discussion, we note that in the case of protein kinases like ULK3, their enhanced activity in cancer can easily go undetected in global gene expression studies, and that specifically dedicated studies are necessary. These limitations also apply to large-scale functional genomic studies as indicated in reply to reviewer 2.

Review of the Human Protein Atlas identifies ULK3 with low tissue specificity, low overall expression levels in multiple tissues, including cancer of the head and neck, skin and other SCC prone tissues and a favorable prognostic marker in pancreatic cancer (ULK3 protein expression summary - The Human Protein Atlas, <https://www.proteinatlas.org/ENSG00000140474-ULK3>).

Reply: In our manuscript, we show that there is an increased ULK3 expression in SCC cells that can be targeted to suppress their proliferation, but we do not make any claims linking ULK3 expression to late stages clinical outcomes. In fact, we show now that ULK3 levels are also elevated in premalignant skin actinic keratosis lesions (Figure 1 panel e), pointing to an involvement of ULK3 in early steps of cancer development, which could be considered for possible cancer prevention approaches.

Since it has been reported that ULK3 regulates cytokinesis (reference 19) a more plausible explanation for the described phenomena might be that down regulating/ deleting ULK3 inhibits mitosis. As such the paper describes that inhibiting a regulator of mitosis inhibits cell growth, which explains a possible role in tumorigenesis. It would seem more proliferative tissues express more ULK3 which, given its role in mitosis, is somewhat expected.

Reply: We thank the reviewer for the suggestion to consider the involvement of ULK3 in cytokinesis in our system. The impact of ULK3 loss on cytokinesis, as reported in reference 19, is linked to cell stress conditions as: "(ULK3) phosphorylates and binds ESCRT-III subunits via tandem MIT domains, and thereby delays abscission in response to lagging chromosomes, nuclear pore defects, and tension forces at the midbody". Notably, the quoted paper focuses on a cytosolic mitotic checkpoint (measured as midbody formation), the loss of which, by ULK3 loss, should result in a faster rather than decreased rate of proliferation.

In any case, we have directly assessed whether "down regulating/ deleting ULK3 inhibits mitosis", as suggested by the reviewer, by analysis of SCC cells expressing a dual color Fluorescence Ubiquitin Cell Cycle Indicator system (FUCCI) (Sakaue-Sawano et al., Cell 2008). We find that ULK3 silencing in these cells by locked antisense oligonucleotides (LNAs) transfection leads to an increase of cells in G1/S rather than G2/M phases of the cell cycle (Figure 10 panel e). As part of the same analysis, we show that ULK3 silencing does not result in detectable nuclear alterations and/or multinucleated cells as would be expected in the case of a mitotic block (Figure 10 panel f). The results are overall consistent with a separate nuclear ULK3 function at the level of chromatin binding and gene transcription.

Authors seem to imply that ULK3 is a regulator of cell fate but the data supporting this are weak and are most likely a result of regulating mitosis regardless of self-renewal/ differentiation in the context of stem cell fate. Inhibiting keratinocyte proliferation usually results in differentiation since this is the keratinocytes default pathway.

Reply: Previous evidence from a number of laboratories has shown that a block of keratinocyte cell proliferation, such as can be triggered by TGF- β treatment or increased p21^{CDKN1A} expression is not by itself sufficient to trigger differentiation, which instead involves a complex transcription / epigenetic program affecting self-renewal potential and cell fate commitment (Dotto and Rutsgi, Cancer Cell, 2016; Fuchs J Cell Biol, 1990). In this context, we have provided multiple evidence that ULK3 loss in keratinocytes and/or SCC cells impacts key determinants of keratinocyte stem cell potential versus commitment to differentiation, causing specifically :

- i) Degradation of the TP63 protein (Senoo et al., Cell 2006; Truong Genes and Dev. 2006)
- ii) Decreased expression of c-MYC (Gandarillas et al., Genes and Dev, 1997)
- iii) Decreased expression of FOXM1, which has been recently reported to be a key regulator of keratinocyte stem cell phenotype (Enzo et al., Nature Comm, 2021), through direct binding of ULK3 to the promoter and controlling its open chromatin configuration.

In response to the reviewer's concern, we have further investigated the impact of ULK3 loss on HKC and SCC self-renewal potential, utilizing an additional approach of translational significance. Locked antisense oligonucleotides (LNAs) provide an exciting emerging technology for targeting key signaling nodes (eg, Crooke et al., J. Biol. Chem 2021; Roberts et al. Nat. Rev. Drug Discovery 2020). Accordingly, we designed

and tested three different LNAs leading to > 90% downmodulation of ULK3 expression and concomitant TP63 decrease (Figure 10 panels a, b and h). Clonogenicity assays provide a classical method to assess HKC self-renewal potential, which we have complemented by sphere formation assays of single cells plated in matrigel suspension. We show that LNA-mediated downmodulation of ULK3 causes a strong reduction of HKC or SCC self-renewal potential as assessed by both of these assays (Figure 10 panel d).

Use of ULK3 inhibitors (Kasak et al., Characterization of Protein Kinase ULK3 Regulation by Phosphorylation and Inhibition by Small Molecule SU6668. Biochemistry. 2018 Sep 18;57(37):5456-5465. doi: 10.1021/acs.biochem.8b00356. PMID: 30096229) would be of interest in the context of the study.

Reply: We thank the reviewer for the excellent suggestion. As requested, we have examined the effects of the ULK3 inhibitor SU6668 and show, that consistent with the shRNA- and LNA-mediated gene silencing and CRISPR-deletion, treatment of SCC cells with this compound causes : i) cell proliferation ii) colony formation iii) EdU incorporation (Supplementary Figure 2 panels d-f), TP63 levels (Supplementary Figure 3 panel a) in SCC cells and phosphorylation of PRMTs by ULK3 (Figure 7 panel d).

Statements regarding the binding of ULK3 to PRMT1 and 5 are over-interpreting the data (summary) since PLA, Mass spec and IP all identify association/interaction but not direct binding. More detail on the results of the Mass-spec experiment would be helpful; how many peptides identified, how do PRMT1 and 5 sit amongst other proteins identified by this unbiased approach?

Reply: We thank the reviewer for asking for more direct evidence of the ULK3-PRMT association, which we have obtained using baculovirus-expressed recombinant proteins. We show a direct association of ULK3 with both PRMT1 or PRMT5 (Figure 7 panels a and b), which is dependent of native folding of these proteins, i.e. not observed after heat denaturation (Figure 7 panel c). Utilizing this approach, we also show that: i) PRMT1 or PRMT5 are phosphorylated by incubation with ULK3, a reaction which is inhibited by treatment with SU6668 (Figure 7 panel d); ii) ULK3 in turn, is arginine methylated when incubated with PRMT1 or PRMT5 (Supplementary Figure 7 panels c and d).

Regarding the Mass spectrometry analysis, this was only mentioned in the text as part of our initial exploratory studies and, to avoid any confusion, we have decided to remove any mention of it in the resubmission.

All generated datasets should be deposited in public databases, no access was given to GSE183933 for reviewers so it is not clear which generated datasets have been deposited.

Reply: we have double-checked that all the deposited datasets included in the public databases are accessible.

Referencing is rather author centric and key molecular studies on SCC are lacking.

Reply: We apologize for the overseeing, in the resubmission, we included reviews and referenced additional previous studies on SCCs.

Reviewer #2 (Remarks to the Author):

Goruppi et al. introduce ULK3 as a master regulator for SCC cell proliferation and tumorigenesis. The arginine methyltransferases PRMT1 and PRMT5 are identified as ULK3 interactors and substrates. Furthermore, the impact of PRMT activity on chromatin configuration and translation is analyzed. The

authors conclude with suggesting ULK3 and the PRMTs as pharmacologic targets for the treatment of SCCs.

While the findings are novel and convincingly presented, several aspects in the long and detailed line of evidence might need additional experimental support.

Reply: we appreciate the reviewer's positive comments and suggestions for improvement, which we have addressed as specified below.

Comments:

Line 130 The authors describe a novel signaling pathway with ULK3 regulating proliferation, colony formation and tumorigenesis. The mechanism is in place both in SCC cells and HeLa cells. It is worth discussing why this fundamental role of ULK3 as a fitness gene has not been detected in the numerous genome-wide siRNA or CRISPR screens.

Reply: we thank the reviewer for the question. As we now point out in the discussion (page 15 line 11), ULK3 or PRMTs have not been identified in large-scale functional genomic screening assays for fitness or dependency genes in relevant cancer cell types (SCC or HeLa). While these screens are instrumental in identifying novel or established targets, they are never exhaustive, as they critically depend on the depth and efficiency of the targeting gene libraries and the functional assays and operational criteria that were chosen. In fact, well-known genes involved in negative or positive control of SCC development, as identified by other approaches, are often missed. For instance, a recent gene-deletion screen by CRISPR in 21 oral squamous cell carcinoma cell lines uncovered a new dependency for Hippo, while previously identified dependency genes such as PI3CA, CDK6, NOTCH1, p53 were not found (e.g. Chai et al. eLife 2020;9:e57761). In another CRISPR/cas9 screening for gene dependencies performed on 329 cancer cell lines spanning 30 cancer types, analysis of head and neck or lung squamous cell carcinomas subgroups failed to identify well know nodes such as CDK6, p63 and NOTCH1 (Behan et al., Nature 2019).

Line 138 The arrangement of panels in Fig 2 can be made more consistent. Same color code in the graphs - explain why the respective cell lines have been chosen - either show the western blots or omit. Maybe combine panels d to j?

Reply: The color code for indication of the various cell lines in Figure 2 and throughout the manuscript has been made consistent. The cell lines used for these studies were chosen as well characterized representatives of skin and oral SCCs as specified in the result section (page 5 line 20). For an adequate representation of the additional information requested by the reviewers, we have moved panels d to j of the previous Figure 2 to the new Figure 3 (now panels a, c, e, f and g).

Line 154 Please discuss whether or not this result is expected when set in relation to the ULK3 literature.

Reply: we now indicate in the results text (page 6 line 16), that ULK3 silencing results in down-modulation of GLI1 and GLI2 family members is in agreement with what is known of ULK3 as a regulator of Gli activity (through Gli2 phosphorylation) in other cellular systems (dermal fibroblasts, Goruppi et al., Cell Reports 2018; NIH3T3 cells, Maloverjan et al., Exp.Cell.Res. 2010)"

Line 163 Please relate the relative expression from the WB quantification (Fig 2d-j, Fig 3e-f) with the relative expression from the MS quantification. Maybe present the MS analysis first - then the WB analysis ?

Line 229 Please compare the identified ULK3 interaction partners to previously published ULK3 interactors.

Reply: Mass spectrometry analysis was only used as part of our initial exploratory studies and we have only qualitative data, which could not be meaningfully related to the results of co-IP / WB analysis. To avoid any confusion, we have decided to omit the mention of the Mass Spec results and confirm instead the findings of ULK3-PRMT1/5 association through the use of recombinant proteins. We thereby show a direct association of ULK3 with both PRMT1 or PRMT5 (Figure 7 panels a and b), which is dependent on native folding of these proteins, i.e. not observed after heat denaturation (Figure 7 panel c). We also show that: i) PRMT1 or PRMT5 are phosphorylated by incubation with ULK3, a reaction which is inhibited by treatment with ULK3 inhibitor SU6668 (Figure 7 panel d); ii) ULK3 in turn, is arginine methylated when incubated with PRMT1 but not PRMT5 (Supplementary Figure 7 panels c and d).

To answer to reviewer's request, we have examined for the presence of the published ULK3 interacting partners in our mass spectrometry assay and neither CHMP1A (Caballe et al., ELife 2015) or GLI2 (Maloverjan et al., 2010) were among the interactors we identified, likely a consequence of the high stringency of ULK3 immunoprecipitating conditions.

Line 236 ULK3 has been introduced as a nuclear protein (Fig 1c). The ULK3:PRMT1/5 speckles, however, are distributed evenly in nucleus and cytoplasm. In addition to showing the PLA co-localization in Fig 5b, it will be good to also show the subcellular distributions of ULK3 and PRMT1/5.

Reply: we thank the reviewer for the observation of the few cytosolic speckles and, as requested, we have examined the subcellular fraction of ULK3 by both biochemical fractionation and immunofluorescence analysis. We find that, in our cellular system, ULK3 is mostly nuclear and that PRMT1 and PRMT5 also have a nuclear localization that is unaffected by ULK3 gene silencing (Figure 8 panels b and c). This does not rule out possible cytosolic functions that will have to be separately pursued.

Line 247 While ULK3 K44H represents a common inactivating point mutation, ULK3 K139R is likely to be catalytically active. Please discuss / explain.

Reply: while K44H is an inactivating mutation in the ATP-binding site of the kinase domain (Caballe et al., ELife 2015), the mutation K139R has been shown to be also inactivating, mapping in the donor/acceptor region of the ULK3 kinase transferase activity (Maloverjan et al., 2010, Caballe et al., ELife 2015 and <https://www.uniprot.org/uniprotkb/Q6PHR2/entry>).

Line 252 The microscopic PLA is too indirect to establish PRMT1/5 as cellular ULK3 substrates in vitro assays. Additional MS analysis of the enriched phosphoproteins is suggested.

Reply: to complement the in vivo findings, we have performed in vitro kinase assays with purified recombinant proteins, showing that both PRMT 1 and 5 are targets of ULK3 activity, with PRMT 1 and 5 phosphorylation being blocked by the addition of SU6668, a small molecular weight ULK3 inhibitor (Pirsoo et al., Biochim Biophys Acta, 2014; Kasak, Biochemistry 2018). We performed a western blot analysis of the kinase reactions with anti PRMT antibodies after their separation on Phos-TAG gels, which identify the phosphorylations by the appearance of gel-retarded band(s). Both PRMT1 and PRMT5 showed a slower migrating form (Figure 7 panel d), while no such change occurred with the incubation of ULK3 with an unrelated protein such as GFP under the same conditions (Figure 7 panel e). Importantly, we further show that, in parallel with their phosphorylation, pre-incubation of PRMT1 or PRMT5 with ULK3 increases the enzymatic activity of these histone methylases towards a known target, the arginine 3 of histone 4 (H4R3), as detected by using commercial antibodies specific for such epigenetic modification (Figure 7 panels f and g).

Line 401 Please discuss the translational significance in more detail since this is also prominently mentioned in the abstract. Are ULK3 or PRMT1/5 inhibitors available? Will it be advantageous to apply PROTACs over inhibitors? What are potential adverse effects when targeting these proteins pharmacologically?

Reply: we thank the reviewer for the excellent question about the translational significance of the findings, which we have addressed experimentally. As suggested by reviewer 1, we have assessed the impact of a small molecular weight inhibitor of ULK3 kinase activity (SU6668; Pirsoo et al., *Biochim Biophys Acta*, 2014; Kasak, *Biochemistry* 2018) on SCC self-renewal capability as assessed by clonogenicity and proliferation assays (Supplementary Figure 2 panel d-f). We have further assessed by clonogenicity assay on SCC cells the specific inhibitors for PRMT1 MS-203 (Eram, *Achs Chem Biol*, 2013) and PRMT5 EPZ015666 (Chan-Penebre *Nature Chem. Biol.*, 1026), (Supplementary Figure 6 panel c)

As an alternative approach that does not suffer from the lack-of-specificity often found in chemical compounds, we have taken advantage of locked antisense oligonucleotides (LNAs), which provide an exciting technology for targeting key signaling nodes (eg Crooke et al., *J. Biol. Chem* 2021; Roberts et al. *Nat. Rev. Drug Discovery* 2020). We designed and validated two different LNAs leading to > 90% downmodulation of ULK3 expression and very effective in suppressing SCC cell proliferation, colony assay, stem potential, cell cycle and tumorigenicity. (Figure 10 panels a-h)

And there were some typos (lines 142 159 202 360 Fig 2g Fig 2j).

Reply: we apologize for the mistakes and have thoroughly checked the text.

Reviewer #3 (Remarks to the Author):

Sandro et. al demonstrated that ULK3 kinase is upregulated in SCCs, and knockdown of ULK3 attenuates the expression of genes related to oncogenic potential and glycolysis. The oncogenic role of ULK3 was validated in orthotopic models of skin cancer. Mechanistically, ULK3 was found to be associated with PRMT1 and PRMT5, and to be required for their function on chromatin. There are quite a few issues need to be addressed.

1. Abstract is muddled. It is not clear how H4R3 regulates chromatin configuration and cell differentiation as they are two forms of H4R3 methylation, symmetrical and asymmetrical, with distinct functions. I suggest a re-write

REPLY: we thank the reviewer for finding the paper of potential interest. As suggested, we have reworded the abstract for better clarity and with an indication of the additional findings that we have obtained

2. In addition to Figure 2D, genomic DNA sequencing was also needed to confirm the knockout of ULK3 in cells.

REPLY: As Ulk3 silencing (by either shRNA or LNA approaches) impairs proliferation of HKCs and SCC cells, single cell clones emerging from CRISPR-mediated deletion of the gene, would be likely to have secondary compensatory mutations. For this reason, we chose an alternative “en masse” approach, which eliminates the issues of individual clone variability, with infection of cells with high titer lentiviruses co-expressing CAS9 with three different ULK3 targeting gRNAs, followed by short-term (3 days) pooled selection. The efficiency and specificity of the gene deletion procedure were assessed by the well-established Surveyor assay (Qiu et. al, *BioTechniques*, 2018), whereby the specific regions surrounding

each targeted exon sequence were amplified by PCR, the amplicons briefly denatured and renatured, and then tested for the presence of mismatched sequences due to the Cas9 cutting and subsequent repair. As shown in Figure 2 panel d, the expected portion of the ULK3 gene was found to be specifically affected in cells targeted by each of the corresponding gRNAs. Importantly, immunoblot analysis showed that the ULK3 protein is absent in ULK3-gene deleted cells as compared to control cells infected with a lentivirus expressing scrambled gRNA sequence (Figure 2 panel e).

3. The author mentioned that $\Delta Np63$ regulates both proliferation and self-renewal in Figure 2F. However, the previous experiments only touched the Growth aspect.

REPLY: as requested, we have further investigated the impact of ULK3 loss on HKC and SCC self-renewal potential, utilizing an additional approach of translational significance. Locked antisense oligonucleotides (LNAs) provide an exciting emerging technology for targeting key signaling nodes (eg Crooke et al., J. Biol. Chem 2021; Roberts et al. Nat. Rev. Drug Discovery 2020). Accordingly, we designed and tested three different LNAs leading to > 90% downmodulation of ULK3 expression and concomitant TP63 decrease (Figure 10 panels a and h). Clonogenicity assays provide a classical method to assess HKC self-renewal potential, which we have complemented by sphere formation assays of single cells plated in matrigel suspension. We show that LNA-mediated downmodulation of ULK3 causes a strong reduction of SCC self-renewal potential as assessed by both of these assays (Figure 10 panels c and d).

4. It might be better to move Figure 2F-2J to Figure 3 as panel A.

REPLY: As requested, we have moved panels f-j from Figure 2 to a new Figure 3 as panels a, c, d, e, f and g. This has allowed us to add the additional data requested by reviewer 1, as indicated in our reply to her/his comments.

*5. The change of $\Delta Np63$, *gli1/2*, *CDKN1A*, *KRT1*, *KRT10*, *INV*, and *FLG* at protein level needs to be shown.*

REPLY: the consequences of ULK3 silencing on TP63 and of CDKN1A protein expression were already assessed by immunoblot analysis of HKC and SCC cells (previous Figure panels 2 d and i, now Figure 3 panels a, b, f and g), with results that we have now complemented by a similar analysis of KRT10 and INV expression (Figure 3 panel e).

We have also examined the changes in TP63 and KRT10 protein expression in vivo by immunofluorescence analysis of tumors formed by in SCC13 and Cal27 cells plus/minus shRNA- or LNA-mediated ULK3 silencing (previously in Figure 8 panels e and f; now Figure 9 panels e and f), (Figure 10 panel h).

6. Line 148-149, "Thus, ULK3 is an essential determinant of HKC and SCC cell proliferative potential that affects other critical regulators of the growth/differentiation of these cells". I don't see enough evidence for a link between ULK3 and differentiation.

REPLY: We have provided evidence that ULK3 shRNA-mediated silencing upregulates the levels of differentiation markers such as KRT1, KRT10, INV and FLG (Figure 3 panels d and e) in two SCC cell lines in vitro and upregulates levels of KRT10 in intradermal SCC tumors in vivo. In the resubmission, we show a similar up-regulation of KRT10 upon ULK3 silencing by LNAs in SCC tumors implanted in vivo (Figure 9 panel e and Figure 10 panel h).

7. Transcriptomic analysis needs to be done with two independent shRNAs to ensure that the target genes identified for ULK3 are reliable.

REPLY: as we have now better indicate in the results text (page 6 line 32 and page 7 line 7), transcriptomic analysis was already performed with keratinocytes and SCC13 cells plus/minus ULK3 silencing by two independent shRNAs. All data have been deposited (GSE183084, GSE183085).

8. Modulation of metabolism-related genes upon ULK3 silencing should be confirmed by RT-qPCR and western blot analysis in HKC cell and SCC cells.

REPLY: The regulation by ULK3 of the metabolism-related genes found in whole genome transcriptomic of HKCs and SCC13 cells and selected for further studies was indeed confirmed by RT-qPCR (for LDHA, PKM, SLC2A1, TIGAR and as a key upstream regulator, FOXM1; (Figure 4 panel e) or western blot analysis (for GLS and PKM; Figure 4 panel f). The main finding of ULK3-dependent regulation was confirmed by further mechanistic studies for three of these genes (LDHA, PKM and FOXM1) (Figure 5 panels e, f and g; Figure 7 panel d).

9. Steady-state metabolomic analysis need to be done to test the effects of ULK3 in SCC028 cells.

REPLY: Steady-state metabolomic analysis was performed with triplicate dishes of SCC13 cells plus/minus ULK3 silencing by two different shRNAs with the raw data results deposited on Metabolights (<https://www.ebi.ac.uk/metabolights/MTBLS5155>). Metabolites with consistent and statistically significant modulation by ULK3 gene silencing (glutathione 4 folds change (log2), $p < 0.0005$; glutamate 6 folds change (log2), $p < 0.0007$, $n(\text{samples/condition})=3$, one-way ANOVA) were included in the analysis shown in Supplementary Figure 4.

10. In addition to steady-state metabolomic analysis, the levels of metabolites, such as glutamate and glutathione, should also be examined.

REPLY: As requested, we have added a graph with the triplicate values of glutamate and glutathione in triplicate dishes of SCC13 cells with ULK3 silencing by two different shRNA versus a control shRNA, with their respective statistical significance (Supplementary Figure 4). In support of the functional significance of these observations, we show that ULK3 silencing in SCC cells increases the dependency of glutamine consumption, also seen as an increase of GLS protein (Figure 4 panel f), a consequence of glutamine adaptive metabolism in SCCs (Momcilovic et al., Cancer Cell 2018).

11. In addition to H3K27ac, more histone markers are required to support the effects of ULK3 knockdown on chromatin openness.

REPLY: As requested, in the revision (Figure 5 panels e, f and g) we have extended our studies on chromatin regulation upon ULK3 knockdown by ChIP analysis with antibodies against H3K27ac in parallel with H3K9ac and RNA Pol II as two additional markers of chromatin activation at promoter regions (Guenter et al., Cell, 2007). We find that all are reduced at the promoter regions of metabolism-related genes under ULK3 control (LDHA, PKM and FOXM1 in cells with ULK3 gene silencing, while their binding to a housekeeping ULK3-independent gene (TUB1A1) is unaffected (Supplementary Figure 5 panel d). The findings are fully consistent with all our other results on the regulation of these genes.

12. Why those 57 genes showed no sign of reduced H3K27ac level when ULK3 was knocked down?

REPLY: we assume that the reviewer refers to the Venn diagram shown in panel D of Figure 4, where, out of 334 ULK3-bound genes, 277 have H3K27ac down modulated upon ULK3 silencing. We thank the

reviewer for the observation, which we now point out in the discussion (page 15 line 15). As a likely explanation, while levels of H3K27 acetylation play an important role in the control of gene transcription, others can be involved, which may also be affected by ULK3 silencing and resulting PRMT1/5 loss, an interesting topic for further studies.

13. What's the change of H3K27ac in TIGAR, PDK2, PDK3, SLC2A1, SLC1A5, SLC38A6 and GLS1 promoter region?

REPLY: we have aligned the H3K27ac binding peaks of ChIP seq analysis with the loci of the requested genes by IGV software and show ULK3 silencing results in a significant downmodulation of H3K27ac at the promoter regions of the TIGAR, PDK2, PDK3, SLC2A1 and SLC38A6, while the H3K27ac peaks for GLS and GLS2 promoter regions were unchanged. These findings, which can be seen through IGV web link included in the Nature Communication Reporting Summary (<https://tinyurl.com/y3fadfom>), are mentioned in the discussion in the context of what was indicated above, that ULK3 silencing and compromised PRMT1/5 function can affect the expression of target genes by modulation of H3K27 acetylation as well as multiple additional mechanisms.

14. ChIP-seq analysis of metabolic genes needs to be validated by RT-qPCR.

REPLY: As requested, we validated the ULK3 ChIP seq results by ChIP analysis of SCC13 cells plus/minus ULK3 silencing with anti ULK3 antibodies, followed by qPCR of two regions of the promoter of the LDHA, PKM and FOXM1 genes, (Figure 5 panels e, f and g).

15. How is the overlap of genes regulated by PRMT1/5 and ULK3? What are the enriched GO terms for genes co-regulated by PRMT1/5 and ULK3?

REPLY: While the transcriptomic analysis of PRMT5 regulated genes is not available, the list of the genes affected by silencing of ULK3 (this paper) and PRMT1 (Bao et al., Dev Cell, 2017) was presented in the TABLE of former Supplementary figure 4 (now supplementary Figure 6 panel b). 63 genes were consistently regulated by shULK3 and siPRMT1 (Pearson's correlation $r=0.34$; $p=0.0065$). We performed the requested gene ontology study (using David, Panther, Gorilla, ShinyGo and STRING software) and likely for the limited number of genes similarly regulated we found no significant enrichments with $FDR < 0.05$.

16. In Figure 5C, PRMT1 seems to bind to ULK3 with higher affinity than PRMT5 when ULK3 was served as the bait. However, when PRMT1 and PRMT5 were served as baits, ULK3 seems to bind equally well with PRMT1 and PRMT5.

REPLY: We thank the reviewer for the observation, noting that the differences observed in western blots after immunoprecipitation of different proteins are of difficult interpretation as they can involve different affinity and avidity of antibodies. We have employed in vitro binding assays to establish that ULK3 directly interacts with PRMT1 and PRMT5 and that its association depends on the correct folding of these other proteins. Further determination of molecular binding affinity and interacting residues will require a separate dedicate study.

17. In Fig5d, it seems that PRMT1 binds to ULK3K139R with higher affinity than WT and K144H, while PRMT5 binds to K144H with the highest affinity. Please clarify.

REPLY: in the absence of molecular studies specifically devoted to this topic, we would be hesitant to make a quantitative argument, as the differences in immunoblot signals may be a consequence of the serial probing and stripping of the membrane rather than a real difference in affinity.

18. The author proposed that PRMTs might be substrates for ULK3, but why not the other way around, namely ULK3 serves as a substrate for PRMTs?

REPLY: we thank the reviewer for the very interesting question. As indicated in a previous reply, we provide direct evidence that PRMT1 and PRMT5 are substrates of ULK3 and that this is associated with increased PRMT1 and PRMT5 activity, as measured by di-methylation of arginine 3 on histone 4 (Figure 7 panels f and g). As the reviewer suggested, we have also assessed the converse possibility that ULK3 in turn, is a PRMT1 and PRMT5. We find that ULK3 is arginine di-methylated by PRMT1 but not PRMT5 both in vitro with recombinant proteins and in cell culture, with overexpressed proteins (Supplementary Figure 7 panels c and d). The mapping and functional significance of these modifications for ULK3 activity will have to be addressed in future studies devoted to this topic.

19. In Figure 5F, it is unclear how could those antibodies used can specifically detect the phosphorylated PRMT1?

REPLY: The results of Figure 5F are based on proximity ligation assays (PLAs), whereby a positive signal is generated only when two different antibodies – oligonucleotide-tagged – are in close proximity of each other. This PLA approach has been used to assess the phosphorylation status of many proteins by combining protein-specific antibodies with antibodies against phosphorylated ser/thr or tyr residues. Specifically, such an approach has already been employed to assess PRMT1 phosphorylation by the CSKN1A kinase (Bao et al., Dev. Cell 2017).

20. The overlap between PRMT- and ULK3-bound genes needs to be shown.

REPLY: To our knowledge, there are no published data of PRMT5 or PRMT1 ChIP-seq analysis in SCC cells, in which our ULK3 chip seq analysis was performed. Comparing different cell types (Bao et al., Dev Cell 2017) in different laboratory settings vs SCC might not be informative.

21. Which regions in PRMT1 and PRMT5 are phosphorylated by ULK3?

REPLY: in the revised manuscript, we provide evidence that both PRMT1 and PRMT5 are direct substrates for ULK3 activity by in vitro incubation assays and show that this is linked with an increase in PRMTs activity (Figure 7 panels d, f and g). Further structural/functional studies are outside the scope of the paper, which focuses on the biological role of ULK3 in SCCs.

22. In Fig 6E, author found that knockdown of ULK3 reduced the global levels of H4R3 dimethylation. Does ULK3 affect PRMT1 methyltransferase activity?

REPLY: We thank the reviewer for the interesting question. As mentioned above, by direct in vitro assays with purified proteins, we show that, in parallel with their increased phosphorylation, ULK3 enhances the enzymatic activity of PRMT1 and PRMT5 activity, as assessed by histone 4 arginine di-methylation (Figure 7 panels d, f and g).

23. Whether knockdown of ULK3 affects PRMT1 and 5 binding to those metabolic genes?

REPLY: as asked, by chromatin immunoprecipitation assays, we show that silencing of ULK3 in SCC13 cells decreases the association of PRMT1 and PRMT5 to PKM and LDHA promoters (Figure 8 panel d).

24. Whether mutation of the phosphorylation sites in PRMT1 and 5 could affect their chromatin binding needs to be tested?

REPLY: As answered at point 21, further structural/functional studies on the connection between PRMT phosphorylation and enzymatic activity are outside the scope of the paper, which is focused on the biological role of ULK3 in SCCs

25. Effects of PRMT1 and PRMT5 on metabolism should be tested by steady-state metabolomic analysis.

REPLY: The paper is focused on the role of ULK3 in SCCs with the modulation of PRMTs as an underlying mechanism. Direct studies on the function of PRMTs in metabolism, although interesting, are out of scope.

26. Whether PRMT1 and PRMT5 downregulation affects SCC growth in vivo needs to be tested?

REPLY: The paper is focused on the role of ULK3 in SCCs, with its impact on PRMT1/5 recruitment to chromatin as one possible mechanism. We note that the role of PRMT1 (eg Zhao et al, Cell Death & Disease 2019; Zhou et al., Tumor Biology 2017) and PRMT5 (eg Fan et al., Neoplasia 2020; Chen et al., Front. Bioeng. Biot. 2021) on HKC or SCC cell growth have been previously investigated. In reply to the reviewer's question, we have expanded our study by use of specific inhibitors of PRMT1 (inhibitor MS203, Eram et al., ACS Chem Biol, 2016) and PRMT5 (inhibitor EPZ015666, Cahn-Penebre et al., Nat Chem Biol, 2015). We show that these compounds exert inhibitory effects on SCC self-renewal potential similar to those of ULK3 silencing or gene deletion (Supplementary Figure 6 panel c).

REVIEWERS' COMMENTS

Reviewer #1 (Remarks to the Author):

Some aspects of the manuscript have been improved but for this reviewer the overarching message is still not supported by the data.

I am still not convinced by the "direct binding" statement. It is clear that ULK3 associates with and phosphorylates PRMT1 and 5 but direct binding has not been demonstrated since we do not know whether scaffold proteins are involved in the assays.

The abstract has improved (ignoring the direct binding statement) but the title and introduction stating that "ULK3 is an essential determinant of keratinocyte self-renewal and oncogenic potential/tumorigenesis" is not supported by the data. The data support a role for ULK3, are certainly novel and of interest, but a key determinant is a stretch since overall there is a blunting of proliferation and clonogenicity in the absence of ULK3 but ULK3 is not a requirement.

I am not convinced by the rebuttal comments with regards to Figure 1a; Campbell are analyzing TCGA head and neck SCC (522 samples) which I would assume is the same cohort analyzed by Goruppi et al so there is a disconnect between what is significant and what isn't. I have to assume because there is not an explicit reference other than TCGA for the samples analyzed. the full 522 sample set emerged shortly after the publication of the initial 279 data freeze in 2015 so it is confusing where the 2016/2018 update is coming from.

Reviewer #2 (Remarks to the Author):

Extensively revised - I think good to be published.

Best wishes!

Reviewer #3 (Remarks to the Author):

The authors have adequately addressed all my concerns.

Point-by-point answers to the reviewer's comments

Reviewer #1 (Remarks to the Author)

Question #1: I am still not convinced by the "direct binding" statement. It is clear that ULK3 associates with and phosphorylates PRMT1 and 5 but direct binding has not been demonstrated since we do not know whether scaffold proteins are involved in the assays.

Answer: In the revised paper, we dedicated two entire figures (Fig. 7 and Suppl. Fig. 7) to show the direct physical association and reciprocal biochemical modifications by using the recombinant and purified ULK3, PRMT 1 and PRMT5 proteins. These experiments are made in the absence of any other proteins and/or cell extracts and therefore, no scaffold proteins are required for their interaction. We do not rule out their participation in macromolecular complexes with other chromatin proteins with scaffold function as a very likely possibility in intact cells, as we have pointed out in the discussion.

Question #2: The abstract has improved (ignoring the direct binding statement) but the title and introduction stating that "ULK3 is an essential determinant of keratinocyte self-renewal and oncogenic potential/tumorigenesis" is not supported by the data. The data support a role for ULK3, are certainly novel and of interest, but a key determinant is a stretch since overall there is a blunting of proliferation and clonogenicity in the absence of ULK3 but ULK3 is not a requirement.

Answer: To address the reviewer's concern, we will remove the word "essential" from the introduction, stating that "ULK3 is an ~~essential~~ determinant of keratinocyte self-renewal and oncogenic potential/tumorigenesis", and remove the word "key" from the title as well. Our findings and conclusions are fully supported by the gene silencing and deletion studies as well as chemical inhibition and LNA-mediated suppression.

Question #3: I am not convinced by the rebuttal comments with regards to Figure 1a; Campbell are analyzing TCGA head and neck SCC (522 samples) which I would assume is the same cohort analyzed by Goruppi et al so there is a disconnect between what is significant and what isn't. I have to assume because there is not an explicit reference other than TCGA for the samples analyzed. the full 522 sample set emerged shortly after the publication of the initial 279 data freeze in 2015 so it is confusing where the 2016/2018 update is coming from.

Answer: To address the reviewer's previous request to consider the paper by Campbell et al. (Cell Report 2018, submitted April 2017), we have utilized the TCGA data sets on which that study was based, updated to 2016 as we have stated in the Figure 1a legend. We provide the box plots for the statistically significant, which included BLCA (Nature, 2014), LUSC (Nature, 2012) and CESC (Nature 2016), as indicated in the Figure legend. For the Head and Neck samples, the paper indicates in the materials and methods that a total of "522" were used, using as a reference a Cancer Atlas study of 2015 (*Nature. 2015; 517:576–582. [PubMed: 25631445]*). We used this set which, as the reviewer indicates, contains only 279 Head and Neck SCC samples, and the resulting analysis being not significant, was not included. We have added to the methods section the specifics of the composition of all the datasets we analyzed in the paper.